# Covariance adaptive best arm identification

**El Mehdi Saad**
Université Paris-Sacalay, CentraleSupéléc
Laboratoire des Signaux et Systèmes
Paris, France
el-mehdi.saad@centralesupelec.fr

**Gilles Blanchard**
Institut de Mathématique d'Orsay
Université Paris-Saclay
Paris, France
gilles.blanchard@universite-paris-saclay.fr

**Nicolas Verzelen**
Mistea, INRAE
Institut Agro, Université de Montpellier
Montpellier, France
nicolas.verzelen@inrae.fr

## Abstract

We consider the problem of best arm identification in the multi-armed bandit model, under fixed confidence. Given a confidence input $\delta$, the goal is to identify the arm with the highest mean reward with a probability of at least $1 - \delta$, while minimizing the number of arm pulls. While the literature provides solutions to this problem under the assumption of independent arms distributions, we propose a more flexible scenario where arms can be dependent and rewards can be sampled simultaneously. This framework allows the learner to estimate the covariance among the arms distributions, enabling a more efficient identification of the best arm. The relaxed setting we propose is relevant in various applications, such as clinical trials, where similarities between patients or drugs suggest underlying correlations in the outcomes. We introduce new algorithms that adapt to the unknown covariance of the arms and demonstrate through theoretical guarantees that substantial improvement can be achieved over the standard setting. Additionally, we provide new lower bounds for the relaxed setting and present numerical simulations that support their theoretical findings.

## 1 Introduction and setting

Best arm identification (BAI) is a sequential learning and decision problem that refers to finding the arm with the largest mean (average reward) among a finite number of arms in a stochastic multi-armed bandit (MAB) setting. An MAB model $\nu$ is a set of $K$ distributions in $\mathbb{R}$: $\nu_1, \ldots, \nu_K$, with means $\mu_1, \ldots, \mu_K$. An "arm" is identified with the corresponding distribution index. The observation consists in sequential draws ("queries" or "arm pulls") from these distributions, and each such outcome is a "reward". The learner's goal is to identify the optimal arm $i^* := \operatorname{Arg\,Max}_{i \in [\![K]\!]} \mu_i$, efficiently. There are two main variants of BAI problems: The *fixed budget setting* [1, 8], where given a fixed number of queries $T$, the learner allocates queries to candidates arms and provides a

guess for the optimal arm. The theoretical guarantee in this case takes the form of an upper bound on the probability $p_T$ of selecting a sub-optimal arm. The second variant is the *fixed confidence* setting [14, 20], where a confidence parameter $\delta \in (0, 1)$ is given as an input to the learner, and the objective is to output an arm $\psi \in [\![K]\!]$, such that $\mathbb{P}(\psi = i^*) \geq 1 - \delta$, using the least number of arm pulls . The complexity in this case corresponds to the total number of queries made before the algorithm stops and gives a guess for the best arm that is valid with probability at least $1 - \delta$ according to a specified stopping rule. In this paper we specifically focus on the fixed confidence setting.

The problem of best-arm identification with fixed confidence was extensively studied and is well understood in the literature [16, 19, 20, 14]. However, in these previous works, the problem was considered under the assumption that all observed rewards are independent. More precisely, in each round $t$, a fresh sample (reward vector), independent of the past, $(X_{1,t}, \ldots, X_{K,t})$ is secretly drawn from $\nu$ by the environment, and the learner is only allowed to choose one arm $A_t$ out of $K$ (and observe its reward $X_{A_t,t}$). We relax this setting by allowing simultaneous queries. Specifically, we consider the MAB model $\nu$ as a *joint probability distribution* of the $K$ arms, and in each round $t$ the learner chooses a subset $C_t \subset [\![K]\!]$ and observes the rewards $(X_{i,t})_{i \in C_t}$ (see the Game Protocol 1). The high-level idea of our work is that allowing multiple queries per round opens up opportunities to estimate and leverage the underlying structure of the arms distribution, which would otherwise remain inaccessible with one-point feedback. This includes estimating the covariance between rewards at the same time point. It is important to note that our proposed algorithms do not require any prior knowledge of the covariance between arms.

Throughout this paper, we consider two cases: bounded rewards (in Section 4) and Gaussian rewards (in Section 5), with the following assumptions:

**Assumption 1 .** *Suppose that:*

- *IID assumption with respect to $t$: $(X_t)_{t \geq 1} = (X_{1,t}, \ldots, X_{K,t})_{t \geq 1}$ are independent and identically distributed variables following $\nu$.*
- *There is only one optimal arm: $\left| \operatorname{Arg\,Max}_{i \in [\![K]\!]} \mu_i \right| = 1$.*

**Assumption 2 $-\mathcal{B}$.** *Bounded rewards: The support of $\nu$ is in $[0, 1]^K$.*

**Assumption 2 $-\mathcal{G}$.** *Gaussian rewards: $\nu$ is a multivariate normal distribution.*

---

**Protocol 1** The Game Protocol

---

    **Parameters**: $\delta$.
    **while [condition] do**
        Choose a subset $C \subseteq [\![K]\!]$.
        The environment reveals the rewards $(X_i)_{i \in C}$.
    **end while**
    Output the selected arm: $\psi$.

---

A round corresponds to an iteration in Protocol 1. Denote by $i^* \in [\![K]\!]$ the optimal arm. The learner algorithm consists of: a sequence of queried subsets $(C_t)_t$ of $[\![K]\!]$, such that subset $C_t$ at round $t$ is chosen based on past observations, a halting condition to stop sampling (i.e. a stopping time written $\tau$) and an arm $\psi$ to output after stopping the sampling procedure. The theoretical guarantees take the form of a high probability upper-bound on the total number $N$ of queries made through the game:

$$N := \sum_{t=1}^{\tau} |C_t|. \tag{1}$$

## 2 Motivation and main contributions

The query complexity of best arm identification with independent rewards, when arms distributions are $\sigma$-sub-Gaussian is characterized by the following quantity:

$$H(\nu) := \sum_{i \in [\![K]\!] \setminus \{i^*\}} \frac{\sigma^2}{(\mu_{i^*} - \mu_i)^2}. \tag{2}$$

Some $\delta$-PAC algorithms [16] guarantee a total number of queries satisfying $\tau = \tilde{\mathcal{O}}(H(\nu) \log(1/\delta))$ (where $\tilde{\mathcal{O}}(.)$ hides a logarithmic factor in the problem parameters).

Recall that the number of queries required for comparing the means of two variables, namely $X_1 \sim \mathcal{N}(\mu_1, 1)$ and $X_2 \sim \mathcal{N}(\mu_2, 1)$, within a tolerance error $\delta$, is approximately $\mathcal{O}\big(\log(1/\delta)/(\mu_1 - \mu_2)^2\big)$. An alternative way to understand the sample complexity stated in equation (2) is to view the task of identifying the optimal arm $i^*$ as synonymous with determining that the remaining arms $[\![K]\!] \setminus \{i^*\}$ are suboptimal. Consequently, the cost of identifying the best arm corresponds to the sum of the comparison costs between each suboptimal arm and the optimal one.

In practical settings, the arms distributions are not independent. This lack of independence can arise in various scenarios, such as clinical trials, where the effects of drugs on patients with similar traits or comparing drugs with similar components may exhibit underlying correlations. These correlations provide additional information that can potentially expedite the decision-making process [18].

In such cases, Protocol 1 allows the player to estimate the means and the covariances of arms. This additional information naturally raises the following question:

*Can we accelerate best arm identification by leveraging*
*the (unknown) covariance between the arms?*

We give two arguments for a positive answer to the last question: When allowed simultaneous queries (more than one arm per round as in Protocol 1), the learner can adapt to the covariance between variables. To illustrate, consider the following toy example for 2 arms comparison: $X_1 \sim \mathcal{N}(\mu_1, 1)$ and $X_2 = X_1 + Y$ where $Y \sim \epsilon \mathcal{N}(1, 1)$ for some $\epsilon > 0$. BAI algorithms in one query per round framework require $\mathcal{O}((1 + \epsilon)^2 \log(1/\delta)/\epsilon^2)$ queries, which gives $\mathcal{O}(\log(1/\delta)/\epsilon^2)$ for small $\epsilon$. In contrast, when two queries per round are possible, the learner can perform the following test $\mathcal{H}_0$ : "$\mathbb{E}[X_1 - X_2] > 0$" against $\mathcal{H}_1$ : "$\mathbb{E}[X_1 - X_2] \leq 0$". Therefore using standard test algorithms adaptive to unknown variances, such as Student's $t$-test, leads to a number of queries of the order $\mathcal{O}(\mathrm{Var}(X_1 - X_2) \log(1/\delta)/\epsilon^2) = \mathcal{O}(\log(1/\delta))$. Hence a substantial improvement in the sample complexity can be achieved when leveraging the covariance. We can also go one step further to even reduce the sample complexity in some settings. Indeed, we can establish the sub-optimality of an arm $i$ by comparing it with another sub-optimal arm $j$ faster than when comparing it to the optimal arm $i^*$. To illustrate consider the following toy example with $K = 3$: let $X_1 \sim \mathcal{N}(\mu_1, 1)$, $X_2 = X_1 + Y$ where $Y \sim \epsilon \mathcal{N}(1, 1)$ and $X_3 \sim \mathcal{N}(\mu_1 + 2\epsilon, 1)$, with $X_1, Y$ and $X_3$ independent. $X_3$ is clearly the optimal arm. Eliminating $X_1$ with a comparison with $X_3$ requires $\mathcal{O}(\log(1/\delta)/\epsilon^2)$ queries, while comparing $X_1$ with the sub-optimal arm $X_2$ requires $\mathcal{O}(\mathrm{Var}(X_1 - X_2) \log(1/\delta)/\epsilon^2) = \mathcal{O}(\log(1/\delta))$.

The aforementioned arguments can be adapted to accommodate bounded variables, such as when comparing correlated Bernoulli variables. These arguments suggest that by utilizing multiple queries per round (as shown in Protocol 1), we can expedite the identification of the best arm compared to the standard one-query-per-setting approach. Specifically, for variables bounded by 1, our algorithm ensures that the cost of eliminating a sub-optimal arm $i \in [\![K]\!]$ is given by:

$$\min_{j \in [\![K]\!]:\, \mu_j > \mu_i} \left\{ \frac{\mathrm{Var}(X_j - X_i)}{(\mu_j - \mu_i)^2} + \frac{1}{\mu_j - \mu_i} \right\}. \tag{3}$$

The additional term $1/(\mu_j - \mu_i)$ arises due to the sub-exponential tail behavior of the sum of bounded variables. To compare the quantity in (3) with its counterpart in the independent case, it is important to note that the minimum is taken over a set that includes the best arm, and that $\mathrm{Var}(X_{i^*} - X_i) \leq 2(\mathrm{Var}(X_{i^*}) + \mathrm{Var}(X_i))$. Consequently, when the variables are bounded, the quantity (3) is no larger than a numerical constant times $H(\nu)$ (its independent case counterpart in 2), with potentially a significant improvements if there is positive correlation between an arm $j \in [\![K]\!]$ with a higher mean and arm $i$ (so that $\mathrm{Var}(X_j - X_i)$ is small). As a result, our algorithm for bounded variables has a sample complexity, up to a logarithmic factor, that corresponds to the sum of quantities (3) over all sub-optimal arms.

We expand our analysis to encompass the scenario where arms follow a Gaussian distribution. In this context, our procedure ensures that the cost of eliminating a sub-optimal arm $i \in [\![K]\!]$ is given by:

$$\min_{\substack{j \in [\![K]\!]:\\ \mu_j > \mu_i}} \left\{ \frac{\mathrm{Var}(X_j - X_i)}{(\mu_j - \mu_i)^2} \vee 1 \right\}.$$

Similar to the setting with bounded variables, the aforementioned quantity is always smaller than its counterpart for independent arms when all variables $X_j$ have a unit variance.

In Section 6, we present two lower bound results for the sample complexity of best arm identification in our multiple query setting, specifically when the arms are correlated. Notably, these lower bounds pinpoint how the optimal sample complexity may decrease with the covariance between arms, making them the first of their kind in the context of best arm identification, to the best of our knowledge. The presented lower bounds are sharp, up to a logarithmic factor, in the case where all arms are positively correlated with the optimal arm. However, it remains an open question to determine a sharper lower bound applicable to a more general class of distributions with arbitrary covariance matrices.

In Section B of the appendix, we introduce a new algorithm that differs from the previous ones by performing comparisons between the candidate arm and a convex combination of the remaining arms, rather than pairwise comparisons. We provide theoretical guarantees for the resulting algorithm.

Lastly, we conduct numerical experiments using synthetic data to assess the practical relevance of our approach.

## 3   Related work

**Best arm identification:**   BAI in the fixed confidence setting was studied by [11], [22], and [12], where the objective is to find $\epsilon$-optimal arms under the PAC ("probably approximately correct") model. A summary of various optimal bounds for this problem is presented in [8, 20]. Prior works on fixed confidence BAI [25] and [17] developed strategies adaptive to the unknown variances of the arms. In contrast, our proposed algorithms demonstrate adaptability to all entries of the covariance matrix of the arms. In particular, we establish that in the worst-case scenario, where the arms are independent, our guarantees align with the guarantees provided by these previous approaches.

**Covariance in the Multi-Armed Bandits model:**   Recently, the concept of leveraging arm dependencies in the multi-armed bandit (MAB) model for best-arm identification was explored in [15]. However, their framework heavily relies on prior knowledge, specifically upper bounds on the conditional expectation of rewards from unobserved arms given the chosen arm's reward. In a similar vein, a game protocol that allows simultaneous queries was examined in [21]. However, their objective differs from ours as their focus is on identifying the most correlated arms, whereas our primary goal is to identify the arm with the highest mean reward.

The extension of the standard multi-armed bandit setting to multiple-point bandit feedback was also considered in the stochastic combinatorial semi-bandit problem (2, 9, 10 and 13). At each round $t \geq 1$, the learner pulls $m$ out of $K$ arms and receives the sum of the pulled arms rewards. The objective is to minimize the cumulative regret with respect to the best choice of arms subset. [27] proposed an algorithm that adapts to the covariance of the covariates within the same arm. While this line of research shares the intuition of exploiting the covariance structure with our paper, there are essential differences between the two settings. In the combinatorial semi-bandit problem, the learner receives the sum of rewards from all selected arms in each round and aims to minimize cumulative regret, necessitating careful exploration during the game. In contrast, our approach does not impose any constraint on the number of queried arms per round, and our focus is purely on exploration.

Simultaneous queries of multiple arms was also considered in the context of graph-based bandit problems [7]. However, in these studies, it is assumed that the distributions of arms are independent.

**Model selection racing:**   Racing algorithms for model selection refers to the problem of selecting the best model out of a finite set efficiently. The main idea consists of early elimination of poorly performing models and concentrating the selection effort on good models. This idea was seemingly first exploited in [23] through Hoeffding Racing. It consists of sequentially constructing a confidence interval for the generalization error of each (non-eliminated) model. Once two intervals become disjoint, the corresponding sub-optimal model is discarded. Later [25] presented an adaptive stopping algorithm using confidence regions derived with empirical Bernstein concentration inequality (3, 24). The resulting algorithm is adaptive to the unknown marginal variances of the models. Similarly, [4] presented a procedure centered around sequential hypothesis testing to make decisions between two possibilities. In their setup, they assume independent samples and use Bernstein concentration inequality tailored for bounded variables to adapt to the variances of the two variables under consideration.

While the idea of exploiting the possible dependence between models was shown [6, 26] to empirically outperform methods based on individual performance monitoring, there is an apparent lack of theoretical guarantees. This work aims to develop a control on the number of sufficient queries for

reliable best arm identification, while being adaptive to the unknown correlation of the candidate arms.

# 4  Algorithm and main theorem for bounded variables

In this section, we focus on the scenario where the arms are bounded by 1. Let us establish the notation we will be using throughout. For each arm $i \in [\![K]\!]$ associated with the reward variable $X_i$, we denote by $\mu_i$ the mean of $X_i$. Additionally, for any pair of arms $i, j \in [\![K]\!]$, we denote $V_{ij}$ as the variance of the difference between $X_i$ and $X_j$, i.e., $V_{ij} := \mathrm{Var}(X_i - X_j)$. We remind the reader that the quantities $V_{ij}$ are unknown to the learner.

Algorithm 2 is developed based on the ideas introduced in Section 2, which involves conducting sequential tests between every pair $(i, j) \in [\![K]\!] \times [\![K]\!]$ of arms. The key element to adapt to the covariance between arms is the utilization of the empirical Bernstein's inequality [24] for the sequence of differences $(X_{i,t} - X_{j,t})_t$ for $i, j \in [\![K]\!]$. To that end, we introduce the following quantity:

$$\hat{\Delta}_{ij}(t, \delta) := \hat{\mu}_{j,t} - \hat{\mu}_{i,t} - \frac{3}{2}\alpha(t, \delta)\sqrt{2\widehat{V}_{ij,t}} - 9\,\alpha^2(t, \delta),$$

where, $\hat{\mu}_{i,t}$ represents the empirical mean of the samples obtained from arm $i$ up to round $t$, and $\widehat{V}_{ij,t}$ denotes the empirical variance associated with the difference variable $(X_i - X_j)$ up to round $t$. The term $\alpha(t, \delta)$ is defined as $\alpha(t, \delta) := \sqrt{\log(1/\delta_t)/(t-1)}$, and $\delta_t$ is given by $\delta/(2K^2 t(t+1))$.

By leveraging the empirical Bernstein's inequality (restated in Theorem K.1), we can establish that if $\hat{\Delta}_{ij}(t, \delta) > 0$ at any time $t$, then with a probability of at least $1 - \delta$, we have $\mu_i < \mu_j$. This observation indicates that arm $i$ is sub-optimal. Furthermore, when $\mu_j > \mu_i$ for $i, j \in [\![K]\!]$, a sufficient sample sizes ensuring that the quantity $\hat{\Delta}_{ij}(t, \delta)$ is positive is proportional to:

$$\Lambda_{ij} := \left(\frac{V_{ij}}{(\mu_i - \mu_j)^2} + \frac{1}{\mu_j - \mu_i}\right) \log\left(\frac{K}{(\mu_j - \mu_i)\delta}\right). \tag{4}$$

Moreover, we show in Lemma E.5 in the appendix that the last quantity is necessary, up to a smaller numerical constant factor, in order to have $\hat{\Delta}_{ij}(t, \delta) > 0$.

Algorithm 2 follows a successive elimination approach based on the tests $\hat{\Delta}_{ij}(t, \delta) > 0$. Our objective is to ensure that any sub-optimal arm $i$ is queried at most $\min_j \Lambda_{ij}$ times, where the minimum is taken over arms with means larger than $\mu_i$. However, this approach poses a challenge: the arm $j^*$ that achieves the minimum of $\Lambda_{ij}$ may be eliminated early in the process, and the algorithm is then constrained to compare arm $i$ with other arms costing larger complexity $\Lambda_{ij}$. To address this limitation, it is useful to continue querying arms even after deciding their sub-optimality through the pairwise tests.

In Algorithm 2, we introduce two sets at round $t$: $S_t$, which contains arms that are candidates to be optimal, initialized as $S_1 = [\![K]\!]$, and $C_t$, which represents the set of arms queried at round $t$. Naturally, $C_t$ contains $S_t$ and also includes arms that were freshly eliminated from $S_t$, as we hope that these arms will help in further eliminating candidate arms from $S_t$ more quickly.

An important consideration is how long the algorithm should continue sampling an arm that has been eliminated from $S_t$. In Theorem E.2, we prove that when arm $j$ is eliminated at round $t$, it is sufficient to keep sampling it up to round $82t$ (the constant 82 is discussed in Remark 1). The rationale behind this number of additional queries is explained in the sketch of the proof of Theorem 4.1. It suggests that if arm $j$ fails to eliminate another arm after round $82t$, then the arm that eliminated $j$ from $S_t$ can ensure faster eliminations for the remaining arms in $S_t$.

**Theorem 4.1.** *Suppose Assumption 1 and $2 - \mathcal{B}$ holds. Consider Algorithm 2, with input $\delta \in (0, 1)$. We have with probability at least $1 - \delta$: the algorithm identifies the best arm and the total number of queries $N$ satisfies:*

$$N \leq c \log(K\Lambda\delta^{-1}) \sum_{i \in [\![K]\!] \setminus \{i^*\}} \min_{\substack{j \in [\![K]\!]: \\ \mu_j > \mu_i}} \left\{\frac{V_{ij}}{(\mu_i - \mu_j)^2} + \frac{1}{\mu_j - \mu_i}\right\}, \tag{5}$$

*where $\Lambda = \max_{i \in [\![K]\!]} \min_{\substack{j \in [\![K]\!]: \\ \mu_j > \mu_i}} \left\{\frac{V_{ij}}{(\mu_i - \mu_j)^2} + \frac{1}{\mu_j - \mu_i}\right\}$ and $c$ is a numerical constant.*

---

**Algorithm 2** Pairwise-BAI

---

1: **Input** $\delta$.
2: **Initialization:**
3:       Query all arms for 2 rounds and compute the empirical means vector $\hat{\boldsymbol{\mu}}_t$, $t \leftarrow 3$.
4:       $S \leftarrow [\![K]\!]$,      /*Set of candidate arms*/
5:       $C \leftarrow [\![K]\!]$,      /*Set of queried arms*/
6: **while** $|S| > 1$ **do**
7:    Jointly query all arms in $C$.
8:    Update $\hat{\boldsymbol{\mu}}_t$ and compute $\max_{j \in C} \hat{\Delta}_{ij}(t, \delta)$ for each $i \in S$.
9:    **for** $i \in S$ **do**
10:       **if** $\max_{j \in C} \hat{\Delta}_{ij}(t, \delta) > 0$ **then**
11:          Eliminate $i$ from $S$: $S \leftarrow S \setminus \{i\}$.      /* $i$ is sub-optimal */
12:          Mark $i$ for elimination from $C$ at round: $82\, t$.
13:       **end if**
14:    **end for**
15:    **Increment** t.
16: **end while**
17: **Return** $S$.

---

*Moreover, if we omit line 12 from Algorithm 2, that is we do not query non-candidate arms, we have with probability at least $1 - \delta$: the algorithm identifies the best arm and the total number of queries $N$ satisfies:*

$$N \leq c \log(K \Lambda \delta^{-1}) \sum_{i \in [\![K]\!] \setminus \{i^*\}} \left\{ \frac{V_{ii^*}}{(\mu_{i^*} - \mu_i)^2} + \frac{1}{\mu_{i^*} - \mu_i} \right\}, \qquad (6)$$

*where $\Lambda$ is defined above and $c$ is a numerical constant.*

**Summary of the proof for bound** (5): Let $i$ denote a sub-optimal arm and $\Upsilon_i := \text{Arg} \text{Min}_j \Lambda_{ij}$. First, we show that at round $t$, if $i \in S_t$ then necessarily we have $\Upsilon_i \cap C_t \neq \emptyset$. We proceed by a contradiction argument: assume $\Upsilon_i \cap C_t = \emptyset$, let $j$ denote the element of $\Upsilon_i$ with the largest mean. Let $k$ denote the arm that eliminated $j$ from $S$ at a round $s < t$. Lemma E.3 shows that, since $\hat{\Delta}_{jk}(s, \delta) > 0$, we necessarily have $\log(1/\delta) \Lambda_{jk}/4 \lesssim s$. Moreover, $j$ was kept up to round $82s$ and in this last round we had $\hat{\Delta}_{ij}(82s, \delta) \leq 0$, which gives by Lemma E.3: $82s \lesssim (25/2) \log(1/\delta_{82s}) \Lambda_{ij}$. Combining the two bounds on $s$, gives: $\Lambda_{jk} < \Lambda_{ij}$. Finally, we use the ultra-metric property satisfied by the quantities $\Lambda_{uv}$ (Lemma E.6), stating that: $\Lambda_{ik} \leq \max\{\Lambda_{ij}, \Lambda_{jk}\}$. Combining the last inequality with the latter, we get $\Lambda_{ik} \leq \Lambda_{ij}$, which means that $k \in \Upsilon_i$. The contradiction arises from the fact that $j$ is the element of the largest mean in $\Upsilon_i$ and arm $k$ eliminated $j$ (hence $\mu_k > \mu_j$). We conclude that necessarily $\Upsilon_i \cap C_t \neq \emptyset$, therefore by Lemma E.3, at a round of the order of $\min_j \Lambda_{ij}$, we will necessarily have $\hat{\Delta}_{ij}(t, \delta) > 0$ for some $j \in C_t$.

**Remark 1.** *From a practical standpoint, to keep sampling an arm eliminated at $t$ for additional $81\, t$ rounds is a conservative approach. The stated value of $81$ is determined by specifics of the proof, but we believe that it can be optimized to a smaller constant based on the insights gained from numerical simulations. Furthermore, even if we omit the oversampling step, as presented in Theorem 4.1, the algorithm is still guaranteed to identify the best arm with probability $1 - \delta$. Only the query complexity guarantee is weaker, but the algorithm may still lead to effective arm elimination, although with potentially slightly slower convergence.*

**Remark 2.** *The idea of successive elimination based on evaluating the differences between variables was previously introduced in [28], in the context of model selection aggregation. Their analysis allows having a bound slightly looser than bound (6) (with the distances between variables instead of variances). On the other hand, Theorem 4.1 in our paper provides a sharper bound in (5).*

First, it is important to note that bound (5) is always sharper (up to a numerical constant factor) than bound (6) because the optimal arm is included in the set over which the minimum in bound (5) is taken. On the other hand, bound (5) can be smaller than bound (6) by a factor of $1/K$. This situation can arise in scenarios where the $K - 1$ sub-optimal arms, which have close means, are highly correlated, while the optimal arm is independent of the rest. In such a situation, the terms in

the sum in (5) corresponding to the correlated sub-optimal arms are relatively small, except for the arm (denoted as $i$) with the largest mean among the $K - 1$ correlated suboptimal rewards. That last remaining arm will be eliminated by the optimal arm, incurring a potentially large cost, but for that arm only. On the other hand, each term in the second bound (6) would be of the order of the last cost.

To provide perspective on our guarantees compared to those developed in the independent one query per round setting, it is worth noting that the standard guarantees in that setting provide a sample complexity corresponding to $\sum_{i \neq i^*} \log(1/\delta)/(\mu_{i^*} - \mu_i)^2$. Since variables are bounded by 1, their variances are bounded by 1. Therefore, in the worst case, we recover the previous guarantees with a numerical constant factor of 2.

A refined adaptive algorithm presented in [25] also utilizes a successive elimination approach using confidence intervals for the arm means based on the empirical Bernstein's inequality. However, unlike our algorithm, they use the concentration inequality to evaluate each arm's mean independently of the other arms. Their approach allows for adaptability to individual arm variances, and is particularly beneficial are small, i.e., $\mathrm{Var}(X_i) \ll 1$. The sample complexity of their algorithm is of the order of:

$$\log(K\Gamma\delta^{-1}) \sum_{i \neq i^*} \frac{\mathrm{Var}(X_i) + \mathrm{Var}(X_{i^*})}{(\mu_{i^*} - \mu_{i^*})^2} \ ,$$

where $\Gamma := \max_{i \neq i^*}(\mathrm{Var}(X_i) + \mathrm{Var}(X_{i^*}))/(\mu_{i^*} - \mu_{i^*})^2$. Neglecting numerical constant factors, the last bound is larger than both our bounds (5) and (6). This is because the variance of the differences can be bounded as follows: $V_{ii^*} \leq 2(\mathrm{Var}(X_i) + \mathrm{Var}(X_{i^*}))$.

# 5 Algorithms and main theorem for Gaussian distributions

In this section, we address the scenario where arms are assumed to follow a Gaussian distribution. We consider a setting where the learner has no prior knowledge about the arms' distribution parameters, and we continue using the notation introduced in the previous section. The main difference between this case and the bounded variables setting, other than the form of the sample complexity obtained, is the behavior of the algorithm when variances of differences between variables tend to 0, displayed by the second bound in Theorem 5.1.

Our algorithm relies on the empirical Bernstein inequality, which was originally designed in the literature for bounded variables [3, 24]. We have extended this inequality to accommodate Gaussian variables by leveraging existing Gaussian concentration results. Note that extending such inequalities more generally for sub-Gaussian variables is a non-trivial task. One possible direction is to suppose that arms follow a sub-Gaussian distribution and satisfy a Bernstein moment assumption (such extensions were pointed by works on bounded variables e.g., **?** ). Given the last class of distributions, we can build on the standard Bernstein inequality with known variance, then plug in an estimate of the empirical variance leveraging the concentration of quadratic forms (see 5). However, it remains uncertain whether an extension for sub-Gaussian variables (without additional assumptions) is practically feasible.

We extend the previous algorithm to the Gaussian case by performing sequential tests between pairs of arms $(i, j) \in [\![K]\!]$. We establish a confidence bound for the difference variables $(X_i - X_j)$ (Lemma D.1 in the appendix) and introduce the following quantity:

$$\hat{\Delta}'_{ij}(t, \delta) := \hat{\mu}_{j,t} - \hat{\mu}_{i,t} - \frac{3}{2}\alpha(t, \delta)\sqrt{2f(\alpha(t, \delta))\,\hat{V}_{ij,t}}, \tag{7}$$

where the $f$ is defined by: $f(x) = \exp(2x + 1)$ if $x \geq 1/3$, and $f(x) = 1/(1 - 2x)$ otherwise.

Using the empirical confidence bounds on the differences between arms, we apply the same procedure as in the case of bounded variables. We make one modification to Algorithm 2: we use the quantities $\hat{\Delta}'_{ij}(t, \delta)$ instead of $\hat{\Delta}_{ij}(t, \delta)$.

By following the same analysis as in the bounded setting, we establish that the sample complexity required for the comparison tests between arms $i$ and $j$ is characterized by the quantity $\frac{V_{ij}}{(\mu_i - \mu_j)^2} \vee 1$. The following theorem provides guarantees on the algorithm presented above:

**Theorem 5.1.** *Suppose Assumptions 1 and 2 $-\mathcal{G}$ holds. Consider the algorithm described above, with input $\delta \in (0, 1)$. We have with probability at least $1 - \delta$: the algorithm identifies the best arm*

*and the total number of queries $N$ satisfies:*

$$N \leq c \log(K\Lambda\delta^{-1}) \sum_{i \in [\![K]\!]\setminus\{i^*\}} \min_{\substack{j \in [\![K]\!]:\\ \mu_j > \mu_i}} \left\{ \frac{V_{ij}}{(\mu_i - \mu_j)^2} \vee 1 \right\}, \tag{8}$$

*where $\Lambda = \max_{i \in [\![K]\!]} \min_{\substack{j \in [\![K]\!]:\\ \mu_j > \mu_i}} \left\{ \frac{V_{ij}}{(\mu_i - \mu_j)^2} \vee 1 \right\}$ and $c$ is a numerical constant.*

*Moreover, if we omit line 12 from Algorithm 2, that is we do not query non-candidate arms, we have with probability at least $1 - \delta$: the algorithm identifies the best arm and the total number of queries $N$ satisfies:*

$$N \leq 3K + c \log\left(\frac{K\delta^{-1}}{\log(1 + 1/\Lambda)}\right) \sum_{i \in [\![K]\!]\setminus\{i^*\}} \frac{1}{\log\left(1 + \frac{(\mu_{i^*} - \mu_i)^2}{V_{ii^*}}\right)}, \tag{9}$$

*where $\Lambda = \max_{i \neq i^*} V_{ii^*}/(\mu_{i^*} - \mu_i)^2$ is defined above and $c$ is a numerical constant.*

**Remarks.** *On the sample complexity cost of being adaptive to the variance: If the sample size is larger than $\log(1/\delta)$, the cost of plugging in the empirical variance estimate into the Chernoff's concentration inequality is only a multiplicative constant slightly larger than one (nearly $1 + 2\sqrt{\log(1/\delta)/n}$). However, in the case of a small sample regime ($n < \log(1/\delta)$), the cost is a multiplicative factor of $\exp(\sqrt{\log(1/\delta)/n} + 1/2)$ due to the nature of the left tail of the chi-squared distribution (see Sections D and K of the appendix for detailed calculations). For most natural regimes, the number of queries made for each arm is larger than $\log(1/\delta)$, hence the last described effect does not arise. However, in some specific regimes (such as the case of very small variances of the arms) an optimal algorithm should query less than $\log(1/\delta)$ samples, which necessitates introducing the exponential multiplicative term above into the concentration upper bound. This translates into a different form of guarantee presented in Theorem 5.1, inequality (9). It is important to note the cost in this regime cannot be avoided as highlighted by our lower bound presented in Theorem 6.2.*

Theorem 5.1 above shows that Algorithm 2 is applied to Gaussian distribution guarantees that each sub-optimal arm $i$ is eliminated after roughly $\min_j V_{ij}^2/(\mu_i - \mu_j)^2$ queries. Bounds (8) and (9) derived from our algorithm are smaller than the standard complexity bound in the independent case, which is given by $\sum_{i \neq i^*} \sigma^2/(\mu_i - \mu_{i^*})^2$ when all arms have variances smaller than $\sigma^2$. It is important to note that unlike most existing procedures in the literature that achieve the standard complexity bound, our algorithm does not require knowledge of the parameter $\sigma^2$.

**About the upper bound** (9): This bound can be further bounded by $\log(\delta^{-1}) \sum_{i \neq i^*} \frac{V_{ii^*}}{(\mu_i - \mu_{i^*})^2} \vee 1$. The form presented in Theorem 5.1 is particularly sharp when the variances $V_{ii^*}$ tend to zero, resulting in a constant upper bound. Recently, in the independent setting where variances are unknown, [17] analyzed the comparison of two arms $i$ and $j$. They derived a complexity of the order $1/\log\left(1 + \frac{(\mu_i - \mu_j)^2}{(\sigma_i^2 + \sigma_j^2)}\right)$, where $\sigma_i^2$ and $\sigma_j^2$ are the variances of arms $i$ and $j$, respectively. This result is reflected in our bound (9), where instead of the sum of variances, our bound considers the variance of the difference, leading to a sharper bound. In Section 6, we provide a lower bound that nearly matches (9).

## 6 Lower bounds

In this section, we present lower bounds for the problem of best arm identification with multiple queries per round, following Protocol 1. We provide lower bounds for both the bounded distributions setting (Theorem 6.1) and the Gaussian distributions setting (Theorem 6.2). It is important to note that our lower bounds are derived considering a class of correlated arm distributions. Therefore, the results obtained in the standard one-query-per-round setting [20] do not hold in our setting.

Our first lower bound is derived for the case where arms follow a Bernoulli distribution. Let $\boldsymbol{\mu} = (\mu_i)_{i \in [\![K]\!]}$ be a sequence of means in $[1/4, 3/4]$, denote by $i^*$ the index of the largest mean. Consider a sequence of positive numbers $\boldsymbol{V} = (V_{ii^*})_{i \in [\![K]\!]}$. We define $\mathbb{B}_K(\boldsymbol{\mu}, \boldsymbol{V})$ as the set of Bernoulli arm distributions such that: (i) $\mathbb{E}[X_i] = \mu_i$; (ii) $\text{Var}(X_i - X_{i^*}) \leq V_{ii^*}$ for all $i \in [\![K]\!]$.

For Bernoulli distributions, the variances and means are linked. Specifically, Lemma K.8 demonstrates that for any pair of Bernoulli random variables $(B_1, B_2)$ with means $(b_1, b_2)$, the following inequality holds: $(b_1 - b_2) - (b_1 - b_2)^2 \leq \mathrm{Var}(B_1 - B_2) \leq \min\{2 - (b_1 + b_2); b_1 + b_2\} - (b_1 - b_2)^2$.

We assume that the sequences $\boldsymbol{V}$ and $\boldsymbol{\mu}$ satisfy the aforementioned condition for $(X_i, X_{i^*})_{i \in \llbracket K \rrbracket}$, as otherwise the class $\mathbb{B}_K(\boldsymbol{\mu}, \boldsymbol{V})$ would be an empty set. Theorem 6.1 provides a lower bound on the sample complexity required for best arm identification in the worst-case scenario over the class $\mathbb{B}_K(\boldsymbol{\mu}, \boldsymbol{V})$.

**Theorem 6.1.** *Let $K \geq 2$ and $\delta \in (0, 1)$. For any $\delta$-sound algorithm, we have*

$$\max_{\mathcal{B} \in \mathbb{B}_K(\boldsymbol{\mu}, \boldsymbol{V})} \mathbb{E}_{\mathcal{B}}[N] \geq \frac{1}{8} \log(1/4\delta) \sum_{i \in \llbracket K \rrbracket \setminus \{i^*\}} \max\left\{ \frac{V_{ii^*}}{(\mu_{i^*} - \mu_i)^2}, \frac{1}{\mu_{i^*} - \mu_i} \right\},$$

*where $N$ is the total number of queries.*

The presented lower bound takes into account the correlation between arms by incorporating the quantities $V_{ii^*}$ as upper bounds for the variances of the differences between arm $i$ and the optimal arm $i^*$. This lower bound indicates that for class $\mathbb{B}_K(\boldsymbol{\mu}, \boldsymbol{V})$, Algorithm 2 is nearly optimal.

Next, we provide a lower bound in the Gaussian case. Let $\boldsymbol{\mu} = (\mu_i)_{i \in \llbracket K \rrbracket}$ be a sequence of means, where $i^*$ denotes the index of the largest mean. We also consider a sequence of positive numbers $\boldsymbol{V} = (V_{ii^*})_{i \in \llbracket K \rrbracket}$. We define $\mathbb{G}_K(\boldsymbol{\mu}, \boldsymbol{V})$ as the set of Gaussian arm distributions satisfying the following conditions: (i) $\mathbb{E}[X_i] = \mu_i$, (ii) $\mathrm{Var}(X_i - X_{i^*}) = V_{ii^*}$, and (iii) $\mathrm{Var}(X_i) \geq 1$ for all $i \in \llbracket K \rrbracket$. Theorem 6.2 provides a lower bound on the sample complexity required for best arm identification in the worst-case scenario over the class $\mathbb{G}_K(\boldsymbol{\mu}, \boldsymbol{V})$.

**Theorem 6.2.** *Let $K \geq 2$ and $\delta \in (0, 1)$. For any $\delta$-sound algorithm, we have*

$$\max_{\mathcal{G} \in \mathbb{G}_K(\boldsymbol{\mu}, \boldsymbol{V})} \mathbb{E}_{\mathcal{G}}[N] \geq 2 \log(1/4\delta) \sum_{i \in \llbracket K \rrbracket \setminus \{i^*\}} \frac{1}{\log\left(1 + \frac{(\mu_i - \mu_{i^*})^2}{V_{ii^*}^2}\right)},$$

*where $N$ is the total number of queries.*

Theorem 6.2 demonstrates that our algorithm achieves near-optimal performance over $\mathbb{G}_K(\boldsymbol{\mu}, \boldsymbol{V})$, up to a logarithmic factor.

## 7 Numerical simulations

We consider the Gaussian rewards scenario. We compare our algorithm Pairwise-BAI (Algorithm 2) to 3 benchmark algorithms: Hoeffding race [23], adapted to the Gaussian setting (consisting of successive elimination based on Chernoff's bounds) and LUCB [19], which is an instantiation of the upper confidence bound (UCB) method. We assume that the last two algorithms have a prior knowledge on the variances of the arms. The third benchmark algorithm consists of using a successive elimination approach using the empirical estimates of the variances. We evaluated two variations of our algorithm. The first one, Pairwise-BAI+, implemented Algorithm 2 for Gaussian variables. In this instance, we modified line 12 by continuing to sample sub-optimal arms that were eliminated at round $t$ until round $2t$ instead of $82t$. We stress that both variants guarantee a $\delta$-sound decision on the optimal arm (see Theorem 5.1). The second instance involved removing the last instruction, meaning we directly stopped querying sub-optimal arms. Figure 1 displays the average sample complexities for each considered algorithm. As expected, the larger the correlation between arms, the better Pairwise-BAI performs.

The second experiment aims to demonstrate that Algorithm 2 adapts to the covariance between sub-optimal arms, as indicated by bound (8) in Theorem 4.1, rather than solely adapting to the correlation with the optimal arm as shown by (9). In this experiment, we consider that the arms are organized into clusters, where each pair of arms within the same cluster exhibits a high correlation (close to 1), while arms from different clusters are independent. According to bound (8) (see also discussion after Theorem 4.1) we expect to observe a gain of up to a factor corresponding to the number of arms per cluster compared to algorithms that do not consider covariance. (Since the total number of arms is kept fixed, the number of arms per cluster scales as the inverse of the number of clusters.)

Figure 1 illustrates the results, showing that the performance of Pairwise-BAI improves as the number of clusters decreases (indicating a larger number of correlated arms). This suggests that increasing

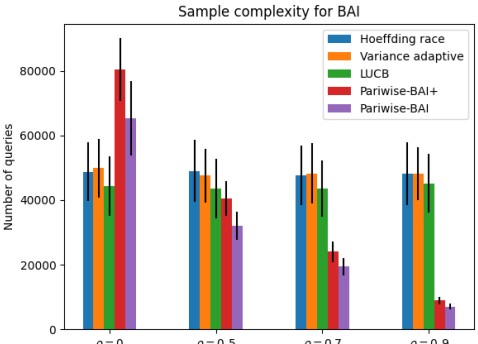
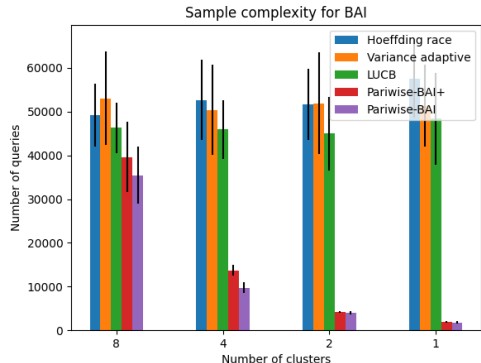

Figure 1: Left figure: the average sample complexities, in 4 scenarios with $K = 10$ arms with means $\mu_i = i/10$. The covariance matrix $C$ is defined as follows: for $i \in [\![K]\!]$, $C_{ii} = 1$ and for $i \neq j$: $C_{ij} = \rho$. We consider 4 scenarios with the correlation: $\rho \in \{0, 0.5, 0.7, 0.9\}$. Right figure: the average sample complexities with $K = 16$ arms with means $\mu_i = i/10$. Arms in the same cluster have a correlation of $0.99$, and arms from different clusters are independent. We consider 4 scenarios with different number of clusters: $n_{\mathrm{cl}} \in \{8, 4, 2, 1\}$. All clusters are of the same size.

the number of correlated sub-optimal arms, which are independent from the optimal arm, still leads to significant performance improvement. These findings support the idea that Pairwise-BAI and Pairwise-BAI+ exhibit behavior that aligns more closely with bound (8) in Theorem 4.1, rather than bound (9).

In both experiments, we observe that Pairwise-BAI+ performs worse compared to Pairwise-BAI, indicating that, empirically, in the given scenarios, continuing to sample sub-optimal arms does not contribute to improved performance. While this modification provides better theoretical guarantees, it may not lead to empirical performance improvements in general scenarios.

# 8 Conclusion and future directions

This work gives rise to several open questions. Firstly, the presented lower bounds take into account partially the covariance between arms. It would be interesting to explore the development of a more precise lower bound that can adapt to any covariance matrix of the arms. Additionally, in terms of the upper bound guarantees, our focus has been on pairwise comparisons, along with an algorithm that compares candidate arms with convex combinations of the remaining arms (Section B). An interesting direction for further research would involve extending this analysis to an intermediate setting, involving comparisons with sparse combinations.

**Acknowledgements:** This work is supported by ANR-21-CE23-0035 (ASCAI) and ANR-19-CHIA-0021-01 (BISCOTTE). This work was conducted while EM Saad was in INRAE Montpellier.

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
