# Supplementary material for:
# Covariance adaptive best arm identification.

## A Notation

- Let $\boldsymbol{X} = (X_1, \ldots, X_K)$ denote the vector of variables associated to the arms.
- Let $\boldsymbol{\mu} = (\mu_i)_{i \in [\![K]\!]}$ denote the vector of arms' means.
- For each $i, j \in [\![K]\!]$, let $V_{ij} = \mathrm{Var}(X_i - X_j)$.
- For each round $t \geq 1$ let $\boldsymbol{X}_t = (X_{1,t}, \ldots, X_{K,t})$ denote the rewards sampled by the environment at round $t$.
- Let $\hat{\mu}_{i,t}$ denote empirical mean of samples pulled from arm $i$ up to round $t$:

$$\hat{\mu}_{i,t} := \frac{1}{t} \sum_{s=1}^{t} X_{i,s}.$$

  Denote $\hat{\boldsymbol{\mu}}_t = (\hat{\mu}_{i,t}, \ldots, \hat{\mu}_{K,t})$.
- Let $(Y_t)_t$ denote a sequence of random variables distributed following $Y$:

$$\widehat{V}_t(Y) = \frac{1}{t(t-1)} \sum_{1 \leq u < v \leq t} (Y_u - Y_v)^2.$$

- For $i, j \in [\![K]\!]$, define the empirical variance for $(X_i - X_j)$ as follows:

$$\widehat{V}_{ij,t} := \frac{1}{t(t-1)} \sum_{1 \leq u < v \leq t} ((X_{i,u} - X_{j,u}) - (X_{i,v} - X_{j,v}))^2. \tag{10}$$

- Define $\delta_t := \delta/(2K^2 t(t+1))$ and $\alpha(t, \delta) := \sqrt{\frac{\log(\delta_t^{-1})}{t-1}}$.
- For bounded variables, define:

$$\hat{\Delta}_{ij}(t, \delta) := \hat{\mu}_{j,t} - \hat{\mu}_{i,t} - \frac{3}{2}\alpha(t, \delta)\sqrt{2\widehat{V}_{ij,t}} - 9\,\alpha^2(t, \delta).$$

- For Gaussian variables, define:

$$\hat{\Delta}'_{ij}(t, \delta) := \hat{\mu}_{j,t} - \hat{\mu}_{i,t} - \frac{3}{2}\alpha(t, \delta)\sqrt{2f(\alpha(t, \delta))\widehat{V}_{ij,t}}.$$

- For bounded variables, define:

$$\hat{\Gamma}_i(w, t, \delta) := \langle w, \hat{\boldsymbol{\mu}}_t \rangle - \hat{\mu}_{i,t} - 2\sqrt{2K\widehat{V}_t(X_i, \langle w, \boldsymbol{X} \rangle)}\alpha(t, \delta) - 14K\|w - e_i\|_1\,\alpha^2(t, \delta),$$

  where $w \in \boldsymbol{B}_1^+$ and $\boldsymbol{B}_1^+ := \{w, w \in [0,1]^K \text{ and } \|w\|_1 = 1\}$.
- Define for $S \subseteq [\![K]\!]$ and $t \geq 1$

$$\boldsymbol{B}_1^+(S) := \{w \in \boldsymbol{B}_1^+ \text{ such that: } \mathrm{supp}(w) \subseteq S\}. \tag{11}$$

- For bounded arms problem, define for $i, j \in [\![K]\!]$:

$$\Lambda_{ij} := \begin{cases} +\infty & \text{if } \mu_j \leq \mu_i \\ \frac{V_{ij}}{(\mu_j - \mu_i)^2} + \frac{3}{\mu_j - \mu_i} & \text{otherwise} \end{cases}$$

- For Gaussian arms problem, define for $i, j \in [\![K]\!]$:

$$\Lambda'_{ij} := \begin{cases} +\infty & \text{if } \mu_j \leq \mu_i \\ \frac{V_{ij}}{(\mu_j - \mu_i)^2} & \text{otherwise} \end{cases}$$

- Observe the quantities $\Lambda_{ij}$ and $\Lambda'_{ij}$ defined in the previous sections are slightly different from the above quantities.

- For bounded arms problem, define for $i \in [\![K]\!]$ and $w \in \boldsymbol{B}_1^+$:

$$\Xi_i(w) := \begin{cases} +\infty & \text{if } \langle w, \boldsymbol{\mu} \rangle \leq \mu_i \\ \max\left\{ \frac{\mathrm{Var}(\langle \boldsymbol{X}, w \rangle - X_i)}{(\langle w, \boldsymbol{\mu} \rangle - \mu_i)^2}, \frac{3\|w - e_i\|_1}{\langle w, \boldsymbol{\mu} \rangle - \mu_i} \right\} & \text{otherwise} \end{cases}$$

- Notation for Algorithms 3 and 2: In round $t$, let $S_t$ denote the set of candidate arms and $C_t$ the set of queried arms at round $t$.

# B  Additional Results for bounded variables: comparison to convex combination

In this section we consider that arms distributions are bounded by 1. We adopt the same notation introduced in the main body and add the following: Let $\boldsymbol{B}_1^+ \subset \mathbb{R}^K$ denote the set of vectors $w$ with non-negative entries such that $\|w\|_1 = 1$. Let $\boldsymbol{X} := (X_1, \ldots, X_K)$ and $\langle .,. \rangle$ denote the euclidean scalar product in $\mathbb{R}^K$. For a subset $A \subset [\![K]\!]$, we denote by $\boldsymbol{B}_1^+(A)$ the set of elements in $\boldsymbol{B}_1^+$ with support in $A$.

While in the previous sections the main idea of the presented procedures is to perform pairwise comparisons between arms, we consider here that for some classes of covariance matrices between the arms, it may is beneficial to perform sequential tests comparing the candidate arms with convex combinations of the non-eliminated arms. For example, for a sub-optimal arm $i$, it is possible to have for some weights vector $w$, with support in $[\![K]\!] \setminus \{i\}$: $\mathrm{Var}(X_i - \langle w, \boldsymbol{X} \rangle) \ll \min_{j \in [\![K]\!]} \mathrm{Var}(X_i - X_j)$. In this case, it is advantageous to eliminate arm $i$ through a comparison with the combination $\langle w, \boldsymbol{X} \rangle$ instead of pairwise comparisons, as concluding that $\mathbb{E}[X_i - \langle w, \boldsymbol{X} \rangle] < 0$ for some $w \in \boldsymbol{B}_1^+$ signifies that arm $i$ is sub-optimal.

The approach used in this section shares similarities with the preceding methodology. More precisely, we develop an empirical second-order concentration inequality over the differences $(X_i - \langle w, \boldsymbol{X} \rangle)$ for $i \in [\![K]\!]$ and $w \in \boldsymbol{B}_1^+$, based on empirical Bernstein inequality and a covering argument over $\boldsymbol{B}_1^+$. We define the following quantity: for $i \in [\![K]\!]$ and $w \in \boldsymbol{B}_1^+$.

$$\hat{\Gamma}_i(w, t, \delta) := \langle w, \hat{\boldsymbol{\mu}}_t \rangle - \hat{\mu}_{i,t} - 2\sqrt{2K\widehat{V}_t(X_i - \langle w, \boldsymbol{X} \rangle)} \alpha(t, \delta) - 14K\|w - e_i\|_1 \alpha^2(t, \delta).$$

Lemma C.2 shows that if $\hat{\Gamma}_i(w, t, \delta) > 0$, then $\mathbb{E}[\langle w, \boldsymbol{X} \rangle] > \mu_i$.

---

**Algorithm 3** Convex-BAI

1: **Input** $\delta$.
2: **Initialization:**
3:       Query all arms for 2 rounds and compute the empirical means vector $\hat{\boldsymbol{\mu}}_t$, $t \leftarrow 3$.
4:       $S \leftarrow [\![K]\!]$,    /*Set of candidate arms*/
5:       $C \leftarrow [\![K]\!]$,    /*Set of queried arms*/
6: **while** $|S| > 1$ **do**
7:     Jointly query all the experts in $C$.
8:     Update $\hat{\boldsymbol{\mu}}_t$ and compute $\sup_{w \in \boldsymbol{B}_1^+(C)} \hat{\Gamma}_i(w, t, \delta)$ for each $i \in S$.
9:     **for** $i \in S$ **do**
10:       **if** $\sup_{w \in \boldsymbol{B}_1^+(C)} \hat{\Gamma}_i(w, t, \delta) > 0$ **then**
11:         Eliminate $i$ from $S$: $S \leftarrow S \setminus \{i\}$.    /* $i$ is sub-optimal */
12:         Activate a counter to eliminate $i$ from $C$ at round: 98 $t$.
13:       **end if**
14:     **end for**
15:     **Increment** $t$.
16: **end while**
17: **Return** $S$.

---

**Remarks.** *In Algorithm 3, we did not specify a method to perform the test:* $\sup_{w \in \boldsymbol{B}_1^+(C_t)} \hat{\Gamma}_i(w, t, \delta) > 0$. *Several developments can be envisioned, such that using methods for convex optimization over a simplex.*

Finally, Theorem B.1 below provides guarantees on the strategy of Algorithm 3. Where tests are performed for each expert against convex combination of all arms.

**Theorem B.1.** *Suppose Assumption 2 $-\mathcal{B}$ hold. Consider Algorithm 3, with input $\delta \in (0, 1)$. With probability at least $1 - \delta$: the procedure selects the best arm $i^*$, and the total number of queries $N$ satisfies:*

$$N \leq c \log(K\Lambda\delta^{-1}) \, K \sum_{i \in [\![K]\!] \setminus \{i^*\}} \min_{\substack{\boldsymbol{w} \in \boldsymbol{B}_1^+: \\ \langle \boldsymbol{w}, \boldsymbol{\mu} \rangle > \mu_i}} \left\{ \frac{\text{Var}(X_i - \langle \boldsymbol{w}, \boldsymbol{X} \rangle)}{(\langle \boldsymbol{w}, \boldsymbol{\mu} \rangle - \mu_i)^2} + \frac{\|\boldsymbol{w} - e_i\|_1}{\langle \boldsymbol{w}, \boldsymbol{\mu} \rangle - \mu_i} \right\}.$$

## C   Concentration lemmas for bounded variables

Define the event $(\mathcal{B}_1)$ for pairwise comparisons:

$\forall t \geq 2, \forall i, j \in [\![K]\!]:$

$$\begin{cases} |(\hat{\mu}_{i,t} - \hat{\mu}_{j,t}) - (\mu_i - \mu_j)| \leq \alpha(t, \delta)\sqrt{2\widehat{V}_{ij,t}} + 6\,\alpha^2(t, \delta) & \text{(12a)} \\ \left| \sqrt{\widehat{V}_{ij,t}} - \sqrt{V_{ij}} \right| \leq 2\sqrt{2}\alpha(t, \delta), & \text{(12b)} \end{cases}$$

where $\widehat{V}_{ij,t}$ is the empirical variance of the sequence $(X_{i,t} - X_{j,t})_{t \geq 1}$, $V_{ij} = \text{Var}(X_i - X_j)$, $\alpha(t, \delta) = \sqrt{\log(\delta_t^{-1})/(t-1)}$ and $\delta_t = 2K^2\delta/(t(t+1))$.

Define the event $(\mathcal{B}_2)$ for comparisons with convex combinations:

$\forall t \geq 2, \forall i \in [\![K]\!], \forall w \in \boldsymbol{B}_1^+(C_t):$

$$\begin{cases} |(\langle w, \hat{\boldsymbol{\mu}_t} \rangle - \hat{\mu}_{i,t}) - (\langle w, \boldsymbol{\mu} \rangle - \mu_i)| \leq \sqrt{2K\widehat{V}_t(X_i - \langle w, \boldsymbol{X} \rangle)}\,\alpha(t, \delta) + 7K\|w - e_i\|_1\alpha^2(t, \delta) & \text{(13a)} \\ \left| \sqrt{\widehat{V}_t(X_i - \langle w, \boldsymbol{X} \rangle)} - \sqrt{\text{Var}(X_i - \langle w, \boldsymbol{X} \rangle)} \right| \leq 3\sqrt{K}\|w - e_i\|_1\,\alpha(t, \delta), & \text{(13b)} \end{cases}$$

where $\boldsymbol{B}_1^+(C_t)$ is defined in Section A (11).

We show that events $(\mathcal{B}_1)$ and $(\mathcal{B}_2)$, defined in (13a), (13b) and (12a), (12b) respectively, hold with high probability.

**Lemma C.1.** *We have $\mathbb{P}(\mathcal{B}_1) \geq 1 - 3\delta$.*

*Proof.* The first inequality is a direct consequence of empirical Bernstein's inequality (Theorem K.1) applied to the sequence of i.i.d variables $(X_{i,s} - X_{j,s})_{s \leq t}$, and using a union bound over $i, j \in [\![K]\!]$ and $t \geq 2$. The second inequality of event $(\mathcal{B}_1)$ is a direct consequence of Theorem K.2.

$\square$

**Lemma C.2.** *We have $\mathbb{P}(\mathcal{B}_2) \geq 1 - 4\delta$.*

*Proof.* Let $\boldsymbol{B}_1$ denote the unit ball with respect to $\|.\|_1$ in $\mathbb{R}^K$. We will show that: $\quad \forall t \geq 2, \forall i \in [\![K]\!], \forall \boldsymbol{u} \in \boldsymbol{B}_1:$

$$\begin{cases} |\langle \boldsymbol{u}, \hat{\boldsymbol{\mu}_t} - \boldsymbol{\mu} \rangle| \leq \sqrt{2K\widehat{V}_t(\langle \boldsymbol{u}, \boldsymbol{X} \rangle)}\,\alpha(t, \delta) + 7K\alpha^2(t, \delta) & \text{(14a)} \\ \left| \sqrt{\widehat{V}_t(\langle \boldsymbol{u}, \boldsymbol{X} \rangle)} - \sqrt{\text{Var}(\langle \boldsymbol{u}, \boldsymbol{X} \rangle)} \right| \leq 3\sqrt{K}\,\alpha(t, \delta), . & \text{(14b)} \end{cases}$$

The result follows by taking $\boldsymbol{u} = (w - e_i)/\|w - e_i\|_1$ or $\boldsymbol{u} = e_i$.

We use a standard covering argument. Recall that the $\epsilon-$covering number for $\boldsymbol{B_1}$, with respect to $\|.\|_1$, is upper bounded by $(3/\epsilon)^K$ (Lemma 5.7 in 31).

Fix $\delta \in (0,1)$. For each $i \in [\![K]\!]$, $t \geq 1$, let $\epsilon_t > 0$ be a parameter to be specified later. Let $\mathcal{N}_t$ be an $\epsilon_t$-cover of the set of $\boldsymbol{B_1}$, with respect to $\|.\|_1$. We will first prove that the event defined in the beginning of the proof is true for all $u \in \mathcal{N}_t$, then using the triangle inequality, we will prove the inequality for any $u \in \boldsymbol{B_1}$.

Let $i \in [\![K]\!]$ and $u \in \mathcal{N}_t$. Applying Theorem K.1 to the sequence of i.i.d variables $(\langle u, \boldsymbol{X}_s \rangle)_{s \leq t}$ bounded by 1 , we have with probability at least $1 - \delta$,

$$|\langle u, \hat{\boldsymbol{\mu}}_t - \boldsymbol{\mu} \rangle| \leq \sqrt{\frac{2\log(3\delta^{-1})\hat{V}_t(\langle u, \boldsymbol{X} \rangle)}{t}} + 6\frac{\log(3\delta^{-1})}{t},$$

where $\hat{V}_t(\langle u, \boldsymbol{X} \rangle)$ denotes the empirical variance of $(\langle u, \boldsymbol{X} \rangle)$ at round $t$. Using a union bound over $t \geq 1$, $i \in [\![K]\!]$ and $u \in \mathcal{N}_t$, we have with probability at least $1 - \delta$: $\forall t \geq 1, i \in [\![K]\!], u \in \mathcal{N}_t$:

$$|\langle u, \hat{\boldsymbol{\mu}}_t - \boldsymbol{\mu} \rangle| \leq \sqrt{2}\sqrt{\frac{\log(|\mathcal{N}_t|\delta_t^{-1})}{t-1}\hat{V}_t(\langle u, \boldsymbol{X} \rangle)} + 6\frac{\log(|\mathcal{N}_t|\delta_t^{-1})}{t-1}$$

$$\leq \alpha(t, \delta/|\mathcal{N}_t|)\sqrt{2\hat{V}_t(\langle u, \boldsymbol{X} \rangle)} + 6\alpha^2(t, \delta/|\mathcal{N}_t|), \tag{15}$$

where $\delta_t = \delta/(2K^2t(t+1))$.

Applying Theorem K.2 to the sequence $(\langle u, \boldsymbol{X} \rangle)$ at round $t$, we have with probability at least $1 - 2\delta$:

$$\left| \sqrt{\hat{V}_t(\langle u, \boldsymbol{X} \rangle)} - \sqrt{\text{Var}(\langle u, \boldsymbol{X} \rangle)} \right| \leq 2\sqrt{\frac{2\log(\delta^{-1})}{t-1}}.$$

Now, we use a union bound over $t \geq 1$, $i \in [\![K]\!]$ and $u \in \mathcal{N}_t$ to obtain with probability at least $1 - \delta$: $\forall t \geq 1, i \in [\![K]\!], u \in \mathcal{N}_t$

$$\left| \sqrt{\hat{V}_t(\langle u, \boldsymbol{X} \rangle)} - \sqrt{\text{Var}(\langle u, \boldsymbol{X} \rangle)} \right| \leq 2\sqrt{2}\alpha(t, \delta/|\mathcal{N}_t|). \tag{16}$$

To wrap up, fix $t \geq 1$ and let $u' \in \boldsymbol{B_1}$. Since $\mathcal{N}_t$ is a covering for $\boldsymbol{B_1}$, we have: $\exists u \in \mathcal{N}_t$ such that $\|u' - u\|_1 \leq \epsilon_t$.

Hence

$$|\langle u', \hat{\boldsymbol{\mu}}_t - \boldsymbol{\mu} \rangle| \leq |\langle u, \hat{\boldsymbol{\mu}}_t - \boldsymbol{\mu} \rangle| + |\langle u - u', \hat{\boldsymbol{\mu}}_t - \boldsymbol{\mu} \rangle|$$

$$\leq \sqrt{2\hat{V}_t(\langle u, \boldsymbol{X} \rangle)}\alpha(t, \delta/|\mathcal{N}_t|) + 6\alpha^2(t, \delta/|\mathcal{N}_t|) + \epsilon_t,$$

where we used (15) and $\|u - u'\|_1 \leq \epsilon_t$. Moreover, we have

$$\sqrt{\hat{V}_t(\langle u, \boldsymbol{X} \rangle)} \leq \sqrt{\hat{V}_t(\langle u', \boldsymbol{X} \rangle)} + \sqrt{\hat{V}_t(\langle u - u', \boldsymbol{X} \rangle)}$$

$$\leq \sqrt{\hat{V}_t(\langle u', \boldsymbol{X} \rangle)} + \|u - u'\|_1$$

$$\leq \sqrt{\hat{V}_t(\langle u', \boldsymbol{X} \rangle)} + \epsilon_t.$$

Therefore

$$|\langle u', \hat{\boldsymbol{\mu}}_t - \boldsymbol{\mu} \rangle| \leq \sqrt{2\hat{V}_t(\langle u', \boldsymbol{X} \rangle)}\, \alpha(t, \delta/|\mathcal{N}_t|)$$

$$+ 6\alpha^2(t, \delta/|\mathcal{N}_t|) + \epsilon_t\Big(1 + \sqrt{2}\alpha(t, \delta/|\mathcal{N}_t|)\Big). \tag{17}$$

We choose $\epsilon_t = 3\delta_t^{\frac{K-1}{K}} = 3\left(\frac{\delta}{2K^2t(t+1)}\right)^{\frac{K-1}{K}}$, therefore

$$
\begin{aligned}
\alpha(t, \delta/|\mathcal{N}_t|) &= \sqrt{\frac{\log\left((3/\epsilon_t)^K \delta_t^{-1}\right)}{t-1}} \\
&= \sqrt{\frac{\log\left(\delta_t^{-(K-1)}\delta_t^{-1}\right)}{t-1}} \\
&= \sqrt{K}\,\alpha(t,\delta).
\end{aligned}
\tag{18}
$$

Furthermore, we have

$$
\begin{aligned}
\epsilon_t\left(1 + \sqrt{2}\alpha(t, \delta/|\mathcal{N}_t|)\right) &\leq 3(\delta_t)^{\frac{K-1}{K}}\left(1 + \sqrt{K}\,\alpha(t,\delta)\right) \\
&\leq 3(\delta_t)^{1/2}\left(1 + \sqrt{K}\,\alpha(t,\delta)\right) \\
&\leq 3\delta^{1/2}\left(\frac{1}{K\log(\delta_t^{-1})} + \frac{2\sqrt{K}}{K\sqrt{(t-1)\log(\delta_t^{-1})}}\right)\frac{\log(\delta_t^{-1})}{t-1} \\
&\leq \alpha^2(t,\delta).
\end{aligned}
\tag{19}
$$

We plug (18) and (19) into (17), and obtain that with probability at least $1 - \delta$

$$
|\langle u', \hat{\boldsymbol{\mu}}_t - \boldsymbol{\mu}\rangle| \leq \sqrt{2K\widehat{V}_t(\langle u', \boldsymbol{X}\rangle)}\alpha(t,\delta) + 7K\alpha^2(t,\delta).
\tag{20}
$$

We proceed similarly for the second concentration inequality. We have with probability at least $1 - \delta$

$$
\begin{aligned}
\left|\sqrt{\widehat{V}_t(\langle u', \boldsymbol{X}\rangle)} - \sqrt{\mathrm{Var}(\langle u', \boldsymbol{X}\rangle)}\right| &\leq \left|\sqrt{\widehat{V}_t(\langle u, \boldsymbol{X}\rangle)} - \sqrt{\mathrm{Var}(\langle u, \boldsymbol{X}\rangle)}\right| + \epsilon_t \\
&\leq 2\sqrt{2}\alpha(t, \delta/|\mathcal{N}_t|) + \epsilon_t \\
&\leq 3\sqrt{K}\alpha(t,\delta).
\end{aligned}
\tag{21}
$$

We conclude by combining (20) and (21). $\qquad\square$

## D   Concentration lemmas for Gaussian variables

Recall the definition $\alpha(t, \delta) = \sqrt{\log(2K^2t(t+1)\delta^{-1})/(t-1)}$. Define the function $f$ for positive numbers as follows:

$$
f(x) := \begin{cases} \exp(2x+1) & \text{if } x \geq 1/3 \\ \frac{1}{1-2x} & \text{otherwise,} \end{cases}
$$

Define the event $(\mathcal{A}_1)$:    $\forall t \geq 2, \forall\, i, j \in [\![K]\!]$ :

$$
\begin{cases} |(\hat{\mu}_{i,t} - \hat{\mu}_{j,t}) - (\mu_i - \mu_j)| \leq \sqrt{2f(\alpha(t,\delta))\widehat{V}_{ij,t}}\,\alpha(t,\delta) & \text{(22a)} \\ \widehat{V}_{ij,t} \leq V_{ij}\left(1 + 2\alpha(t,\delta) + 2\alpha^2(t,\delta)\right) & \text{(22b)} \\ V_{ij} \leq \widehat{V}_{ij,t}\,f(\alpha(t,\delta)). & \text{(22c)} \end{cases}
$$

Lemma below shows that event $(\mathcal{A}_1)$ defined above holds with high probability.

**Lemma D.1.** *We have* $\mathbb{P}(\mathcal{A}_1) \geq 1 - 4\delta$.

*Proof.* We start by proving inequalities (22b) and (22c). We use Lemma K.7 with $Y_t = (X_{i,t} - X_{j,t} - (\mu_i - \mu_j))/V_{ij}$. A union bound over $t \geq 2$ and $i, j \in [\![K]\!]$ gives with probability at least $1 - \delta$

$$
\widehat{V}_{ij,t} \leq V_{ij}\left(1 + 2\alpha(t,\delta) + 2\alpha^2(t,\delta)\right).
$$

For inequality (22c), we apply the first result of Lemma K.7 to the $(Y_t)$ defined above. Using a union bound, we have with probability at least $1 - \delta$

$$\widehat{V}_{ij,t} \geq V_{ij} \max\left\{1 - 2\alpha(t,\delta); e^{-1}\exp\left(-2\alpha^2(t,\delta)\right)\right\}.$$

Inverting the inequality above leads to (22c).

To prove (22a), we use Chernoff's concentration bound for Gaussian variables (and a union bound), with probability at least $1 - \delta$: for all $t \geq 2$ and $i, j \in [\![K]\!]$:

$$|(\hat{\mu}_{i,t} - \hat{\mu}_{j,t}) - (\mu_i - \mu_j)| \leq \sqrt{2V_{ij}}\alpha(t,\delta). \tag{23}$$

We plug-in the bound (22c) to obtain the result.

$\square$

## E    Key lemmas

**Lemma E.1.** *If* $(\mathcal{B}_1)$ *defined in* (12) *holds, we have the following:*

*For any* $i \in [\![K]\!]$, *if there exists* $t \geq 1$ *and* $j \in [\![K]\!]$ *such that* $\hat{\Delta}_{ij}(t,\delta) > 0$, *then* $i \neq i^*$.

*Moreover, if* $(\mathcal{A}_1)$ *defined in* (12) *holds, we have the following:*

*For any* $i \in [\![K]\!]$, *if there exists* $t \geq 2$ *and* $j \in [\![K]\!]$ *such that* $\hat{\Delta}'_{ij}(t,\delta) > 0$, *then* $i \neq i^*$.

*Proof.* Suppose that $(\mathcal{B}_1)$ is true. Let $t \geq 2$, $i, j \in [\![K]\!]$. We have

$$\mu_j - \mu_i = \hat{\Delta}_{ij}(t,\delta) + \mu_j - \mu_i - (\hat{\mu}_{j,t} - \hat{\mu}_{i,t}) + \frac{3}{2}\sqrt{2\widehat{V}_{ij,t}}\alpha(t,\delta) + 9\alpha^2(t,\delta)$$

$$\geq \hat{\Delta}_{ij}(t,\delta),$$

where we used (22a). Finally, if $\hat{\Delta}_{ij}(t,\delta) > 0$, we have $\mu_j > \mu_i$.

Following the exact same steps we have the result for Gaussian variables (the second claim).     $\square$

**Lemma E.2.** *If* $(\mathcal{B}_2)$ *defined in* (13) *holds, we have the following:*

*For any* $i \in [\![K]\!]$, *if there exists* $t \geq 1$ *and* $w \in \boldsymbol{B}_1^+(C_t)$ *such that:* $\hat{\Gamma}_i(w,t,\delta) > 0$, *then* $i \neq i^*$.

*Proof.* Suppose that $(\mathcal{B}_2)$ is true. Let $t \geq 1$, $i \in [\![K]\!]$ and $w \in \boldsymbol{B}_1^+(C_t)$. We have

$$\langle w, \boldsymbol{\mu}\rangle - \mu_i = \hat{\Gamma}_i(w,t,\delta) + \langle w, \boldsymbol{\mu}\rangle - \mu_i - (\langle w, \hat{\boldsymbol{\mu}}_t\rangle - \hat{\mu}_{i,t}) + 2\sqrt{2K\widehat{V}_t(X_i - \langle w, \boldsymbol{X}\rangle)}\,\alpha(t,\delta)$$

$$\qquad\qquad + 14K\|w - e_i\|_1\alpha^2(t,\delta)$$

$$\geq \hat{\Gamma}_i(w,t,\delta),$$

where we used (13a). If $\hat{\Gamma}_i(w,t,\delta) > 0$, we have $\langle w, \boldsymbol{\mu}\rangle > \mu_i$. Since $w$ is a vector of convex weights, we conclude that $\max_{j \in \text{supp}(w)} \mu_j \geq \langle w, \boldsymbol{\mu}\rangle > \mu_i$.     $\square$

**Lemma E.3.** *If* $(\mathcal{B}_1)$ *defined in* (13) *holds, then for any* $t \geq 1$, $i, j \in C_t$: *such that* $\mu_j > \mu_i$

*If* $\hat{\Delta}_{ij}(t,\delta) > 0$, *then*

$$t - 1 \geq \frac{1}{4}\log(\delta_t^{-1})\Lambda_{ij}.$$

*Furthermore, if* $\hat{\Delta}_{ij}(t,\delta) \leq 0$, *then*

$$t - 1 \leq \frac{25}{2}\log(\delta_t^{-1})\Lambda_{ij}.$$

*Proof.* Suppose that $(\mathcal{B}_1)$ is true. Let $t \geq 1$, $i, j \in [\![K]\!]$. Suppose that $\hat{\Delta}_{ij}(t, \delta) > 0$. We have

$$
\begin{aligned}
\mu_j - \mu_i &= \hat{\Delta}_{ij}(t, \delta) - (\hat{\mu}_{j,t} - \hat{\mu}_{i,t}) + \mu_j - \mu_i + \frac{3}{2}\sqrt{2\widehat{V}_{ij,t}}\,\alpha(t, \delta) + 9\alpha^2(t, \delta) \\
&\geq \hat{\Delta}_{ij}(t, \delta) - \sqrt{2V_{ij}}\alpha(t, \delta) - \frac{4}{3}\alpha^2(t, \delta) + \frac{3}{2}\sqrt{2\widehat{V}_{ij,t}}\,\alpha(t, \delta) + 9\alpha^2(t, \delta) \\
&\geq \hat{\Delta}_{ij}(t, \delta) + \sqrt{\frac{V_{ij}}{2}}\,\alpha(t, \delta) + \frac{3}{2}\sqrt{2\widehat{V}_{ij,t}}\alpha(t, \delta) - \frac{3}{2}\sqrt{2V_{ij}}\alpha(t, \delta) + \frac{23}{3}\alpha^2(t, \delta) \\
&> \sqrt{\frac{V_{ij}}{2}}\,\alpha(t, \delta) + \frac{5}{3}\alpha^2(t, \delta), \quad\quad\quad\quad\quad\quad\quad\quad\quad\quad\quad\quad\quad\quad\quad\quad (24)
\end{aligned}
$$

where we used Bennett's inequality (Theorem 3 in [24]) in the second line, (12b) with $\hat{\Delta}_{ij}(t, \delta) > 0$ in the third line.

Solving inequality(24), gives

$$
\begin{aligned}
\alpha(t, \delta) &\leq \frac{3}{14}\left(\sqrt{\frac{V_{ij}}{2} + \frac{20}{3}(\mu_j - \mu_i)} - \sqrt{\frac{V_{ij}}{2}}\right) \\
&= \frac{2(\mu_j - \mu_i)}{\sqrt{\frac{V_{ij}}{2} + \frac{20}{3}(\mu_j - \mu_i)} + \sqrt{\frac{V_{ij}}{2}}}.
\end{aligned}
$$

Therefore, we have

$$
\begin{aligned}
t - 1 &\geq \log(\delta_t^{-1})\left(\frac{V_{ij}}{4(\mu_j - \mu_i)^2} + \frac{5/3}{\mu_j - \mu_i}\right) \\
&\geq \frac{1}{4}\log(\delta_t^{-1})\,\Lambda_{ij}.
\end{aligned}
$$

Which gives the first result.

For the second bound, we proceed similarly. Suppose that $\hat{\Delta}_{ij}(t, \delta) \leq 0$, we have:

$$
\begin{aligned}
\mu_j - \mu_i &= \hat{\Delta}_{ij}(t, \delta_t) - (\hat{\mu}_{j,t} - \hat{\mu}_{i,t}) + \mu_j - \mu_i + \frac{3}{2}\sqrt{2\widehat{V}_{ij,t}}\,\alpha(t, \delta) + 9\alpha^2(t, \delta) \\
&\leq \hat{\Delta}_{ij}(t, \delta_t) + \sqrt{2V_{ij}}\,\alpha(t, \delta) + \frac{4}{3}\alpha^2(t, \delta) + \frac{3}{2}\sqrt{2\widehat{V}_{ij,t}}\,\alpha(t, \delta) + 9\alpha^2(t, \delta) \quad (25) \\
&\leq \hat{\Delta}_{ij}(t, \delta_t) + \frac{5}{2}\sqrt{2V_{ij}}\,\alpha(t, \delta) + \frac{49}{3}\alpha^2(t, \delta) \\
&\leq \frac{5}{2}\sqrt{2V_{ij}}\alpha(t, \delta) + \frac{49}{3}\alpha^2(t, \delta), \quad\quad\quad\quad\quad\quad\quad\quad\quad\quad\quad\quad\quad (26)
\end{aligned}
$$

Similarly to the previous case, we have:

$$
\begin{aligned}
\alpha(t, \delta) &\geq \frac{\sqrt{\frac{25}{2}V_{ij} + \frac{196}{3}(\mu_j - \mu_i)} - \sqrt{\frac{25}{2}V_{ij}}}{98/3} \\
&= \frac{2(\mu_j - \mu_i)}{\sqrt{\frac{25}{2}V_{ij} + \frac{196}{3}(\mu_j - \mu_i)} + \sqrt{\frac{25}{2}V_{ij}}},
\end{aligned}
$$

which gives

$$
t - 1 \leq \log(\delta_t^{-1})\left(\frac{25}{2}\frac{V_{ij}}{(\mu_i - \mu_j)^2} + \frac{98/3}{\mu_j - \mu_i}\right)
$$

We conclude by inverting (26) leading to:

$$
t - 1 \leq \frac{25}{2}\log(\delta_t^{-1})\,\Lambda_{ij}.
$$

$\square$

**Lemma E.4.** *If $(\mathcal{A}_1)$ defined in (13) holds, then for any $t \geq 2$, $i, j \in C_t$, such that $\mu_j > \mu_i$*
*If $\hat{\Delta}'_{ij}(t, \delta) > 0$, then*

$$t - 1 \geq \frac{1}{2} \log(1/\delta_t) \, \Lambda'_{ij}.$$

*Furthermore, if $\hat{\Delta}'_{ij}(t, \delta) \leq 0$, then*

$$t - 1 \leq \frac{25 \log(1/\delta_t)}{\log\left(1 + 1/\Lambda'_{ij}\right)}.$$

*Moreover, if $\hat{\Delta}'_{ij}(t, \delta) \leq 0$, then: if $\Lambda'_{ij} < \frac{1}{5}$*

$$t - 1 \leq \frac{3 \log(1/\delta_t)}{2 \log\left(\frac{\sqrt{2}}{5 \Lambda'_{ij}}\right)},$$

*while if $\Lambda'_{ij} \geq \frac{1}{5}$*

$$t - 1 \leq 25 \log(1/\delta_t) \Lambda'_{ij}.$$

*Proof.* Suppose that $(\mathcal{A}_1)$ is true. Let $t \geq 2$, $i, j \in [\![K]\!]$. Suppose that $\hat{\Delta}'_{ij}(t, \delta_t) > 0$. We have

$$\mu_j - \mu_i = \hat{\Delta}'_{ij}(t, \delta) - (\hat{\mu}_{j,t} - \hat{\mu}_{i,t}) + \mu_j - \mu_i + \frac{3}{2}\sqrt{2f(\alpha(t,\delta))\widehat{V}_{ij,t}} \, \alpha(t,\delta)$$

$$\geq \hat{\Delta}'_{ij}(t, \delta) + \frac{1}{2}\sqrt{2V_{ij}} \, \alpha(t, \delta)$$

$$> \frac{1}{2}\sqrt{2V_{ij}} \, \alpha(t, \delta), \tag{27}$$

where we used (12a) in the second line and (12b) with $\hat{\Delta}'_{ij}(t, \delta_t) > 0$ in the third line.
Therefore inequality(27), gives

$$\alpha(t, \delta) < \frac{\mu_j - \mu_i}{\sqrt{V_{ij}/2}}.$$

Therefore, we have

$$t - 1 \geq \frac{1}{2} \log(1/\delta_t) \frac{V_{ij}}{(\mu_i - \mu_j)^2}$$

$$= \frac{1}{2} \log(1/\delta_t) \, \Lambda'_{ij}$$

Which gives the first result.

For the second bound, suppose that $\hat{\Delta}'_{ij}(t, \delta) \leq 0$, we have:

$$\mu_j - \mu_i = \hat{\Delta}'_{ij}(t, \delta_t) - (\hat{\mu}_{j,t} - \hat{\mu}_{i,t}) + \mu_j - \mu_i + \frac{3}{2}\sqrt{2f(\alpha(t,\delta))\widehat{V}_{ij,t}} \, \alpha(t,\delta)$$

$$\leq \hat{\Delta}'_{ij}(t, \delta_t) + \frac{5}{2}\sqrt{2f(\alpha(t,\delta))\widehat{V}_{ij,t}} \, \alpha(t,\delta)$$

$$\leq \frac{5}{2}\sqrt{V_{ij}}\sqrt{2f(\alpha(t,\delta))(1 + 2\alpha(t,\delta) + 2\alpha^2(t,\delta))} \, \alpha(t,\delta), \tag{28}$$

Hence:

$$f(\alpha(t,\delta))(\alpha^2(t,\delta) + 2\alpha^3(t,\delta) + 2\alpha^4(t,\delta)) \geq \frac{2(\mu_j - \mu_i)^2}{25 V_{ij}} \tag{29}$$

We consider two distinct cases:

**Case 1:** $|\mu_i - \mu_j| > 5\sqrt{V_{ij}}$. Observe that in this case, inequality (29) implies in particular that:

$$f(\alpha(t,\delta))(\alpha^2(t,\delta) + 2\alpha^3(t,\delta) + 2\alpha^4(t,\delta)) \geq 2.$$

Recall that for a positive number $x < 1/3$, we have $(x^2 + 2x^3 + 2x^4)/(1 - 2x^2) < 2$. Hence, for the latter inequality to hold, we necessarily have that: $\alpha(t,\delta) \geq 1/3$. Therefore, using the definition of $f$ we have $f(\alpha(t,\delta)) = \exp(2\alpha^2(t,\delta) + 1)$, taking the logarithms in inequality (29):

$$2\alpha^2(t,\delta) + \log(\alpha^2(t,\delta) + 2\alpha^3(t,\delta) + 2\alpha^4(t,\delta)) \geq 2\log\left(\frac{\sqrt{2}|\mu_i - \mu_j|}{5\sqrt{V_{ij}}}\right),$$

Observe that for any $x > 0$, we have $3x^2 > 2x^2 + \log(x^2 + 2x^3 + 2x^4)$. Therefore

$$\alpha^2(t,\delta) > \frac{2}{3}\log\left(\frac{\sqrt{2}|\mu_i - \mu_j|}{5\sqrt{V_{ij}}}\right).$$

We conclude that

$$t - 1 < \frac{3/2\log(\delta_t^{-1})}{\log\left(\frac{\sqrt{2}|\mu_i - \mu_j|}{5\sqrt{V_{ij}}}\right)} \tag{30}$$

**Case 2:** $|\mu_i - \mu_j| \leq 5\sqrt{V_{ij}}$. If $\alpha(t,\delta) > 1/3$, then

$$t - 1 < 9\log(\delta_t^{-1}).$$

Otherwise, if $\alpha(t,\delta) \leq 1/3$, we have using the definition of $f(\alpha(t,\delta))$ and inequality (29)

$$\frac{2(\mu_j - \mu_i)^2}{25V_{ij}} \leq \alpha^2(t,\delta)\frac{1 + 2\alpha(t,\delta) + 2\alpha^2(t,\delta)}{1 - 2\alpha(t,\delta)}$$

$$< 2\frac{\alpha^2(t,\delta)}{1 - 2\alpha(t,\delta)}.$$

Inverting the inequality above in $t$, we obtain

$$\alpha(t,\delta) \geq \sqrt{\frac{(\mu_j - \mu_i)^2}{25V_{ij}}\left(\frac{(\mu_j - \mu_i)^2}{25V_{ij}} + 1\right)} - \frac{(\mu_j - \mu_i)^2}{25V_{ij}}.$$

Therefore, we have

$$\alpha^2(t,\delta) \geq \frac{(\mu_j - \mu_i)^2}{25V_{ij}}.$$

We conclude that if $|\mu_i - \mu_j| \leq 5\sqrt{V_{ij}}$:

$$t - 1 < 25\log(1/\delta_t)\frac{V_{ij}}{(\mu_i - \mu_j)^2}. \tag{31}$$

Now in order to unify the bounds obtained in Cases 1 and 2, observe that the function $f$ defined for positive numbers by

$$f(x) := \begin{cases} \frac{3}{\log(2/(25x^2))} & \text{if } x \leq 1/5 \\ 25x^2 & \text{otherwise}, \end{cases}$$

satisfies for any $0 < x < 1/5$

$$f(x) \leq \frac{10}{\log(1 + \frac{1}{x^2})},$$

and for $x > 1/5$

$$f(x) \leq \frac{25}{\log(1 + \frac{1}{x^2})},$$

Therefore, we conclude that if $\hat{\Delta}'_{ij}(t,\delta) \leq 0$, then

$$t - 1 \leq \frac{25\log(1/\delta_t)}{\log(1 + 1/\Lambda'_{ij})}.$$

$\square$

**Lemma E.5.** *If $(\mathcal{B}_2)$ defined in (13) holds, then for any $i \in S_t$, $t \geq 1$ and $w \in \boldsymbol{B}_1^+(C_t)$:*

*If $\hat{\Gamma}_i(w, t, \delta) > 0$, then*

$$t \geq K \log(\delta_t^{-1}) \, \Xi_i(w).$$

*Furthermore, if $\hat{\Gamma}_i(w, t, \delta) \leq 0$, then*

$$t \leq 36K \log(\delta_t^{-1}) \, \Xi_i(w).$$

*Proof.* Suppose that $(\mathcal{B}_2)$ is true. Let $t \geq 1$, $i \in S_t$ and $w \in \boldsymbol{B}_1^+(C_t)$. Suppose that $\hat{\Gamma}_i(w, t, \delta) > 0$. We have

$$
\begin{aligned}
\langle w, \boldsymbol{\mu} \rangle - \mu_i &= \hat{\Gamma}_i(w, t, \delta) - (\langle w, \hat{\boldsymbol{\mu}}_t \rangle - \hat{\mu}_{i,t}) + \langle w, \boldsymbol{\mu} \rangle - \mu_i + 2\sqrt{2K\widehat{V}_t(X_i - \langle w, \boldsymbol{X} \rangle)}\,\alpha(t,\delta) \\
&\quad + 14K\|w - e_i\|_1 \alpha^2(t,\delta) \\
&\geq \hat{\Gamma}_i(w, t, \delta) - \sqrt{2K\mathrm{Var}(X_i - \langle w, \boldsymbol{X} \rangle)} - \frac{4}{3}K\alpha^2(t,\delta) + 2\sqrt{2K\widehat{V}_t(X_i - \langle w, \boldsymbol{X} \rangle)}\,\alpha(t,\delta) \\
&\quad + 14K\|w - e_i\|_1 \alpha^2(t,\delta) \\
&\geq \hat{\Gamma}_i(w, t, \delta) + \sqrt{2K\mathrm{Var}(X_i - \langle w, \boldsymbol{X} \rangle)}\,\alpha(t,\delta) + 7K\|w - e_i\|_1 \alpha^2(t,\delta) \\
&> \sqrt{2K\mathrm{Var}(X_i - \langle w, \boldsymbol{X} \rangle)}\,\alpha(t,\delta) + 7K\|w - e_i\|_1 \alpha^2(t,\delta),
\end{aligned}
$$

where we used Bennett's inequality in the second line and (13b) with $\hat{\Gamma}_i(w, t, \delta) > 0$ in the third line and fourth lines.

Solving the inequality above in $\alpha(t, \delta)$, gives

$$
\begin{aligned}
\alpha(t,\delta) &\leq \frac{\sqrt{2K\mathrm{Var}(X_i - \langle w, \boldsymbol{X} \rangle) + 28K\|w - e_i\|_1(\langle w, \boldsymbol{\mu} \rangle - \mu_i)} - \sqrt{2K\mathrm{Var}(X_i - \langle w, \boldsymbol{X} \rangle)}}{14K\|w - e_i\|_1} \\
&= \frac{2(\langle w, \boldsymbol{\mu} \rangle - \mu_i)}{\sqrt{2K\mathrm{Var}(X_i, \langle w, \boldsymbol{X} \rangle) + 28K\|w - e_i\|_1(\langle w, \boldsymbol{\mu} \rangle - \mu_i)} + \sqrt{2K\mathrm{Var}(X_i, \langle w, \boldsymbol{X} \rangle)}}.
\end{aligned}
$$

Therefore, we have

$$
\begin{aligned}
t &\geq K \log(\delta_t^{-1}) \left( \frac{\mathrm{Var}(X_i - \langle w, \boldsymbol{X} \rangle)}{(\langle w, \boldsymbol{\mu} \rangle - \mu_i)^2} + \frac{7\|w - e_i\|_1}{\langle w, \boldsymbol{\mu} \rangle - \mu_i} \right) \\
&\geq K \log(\delta_t^{-1}) \, \Xi_i(w).
\end{aligned}
$$

Which gives the first result.

Now let us prove the second claim. Suppose that $\hat{\Gamma}_i(w, t, \delta) \leq 0$. We have

$$
\begin{aligned}
\langle w, \boldsymbol{\mu} \rangle - \mu_i &= \hat{\Gamma}_i(w, t, \delta) - (\langle w, \hat{\boldsymbol{\mu}}_t \rangle - \hat{\mu}_{i,t}) + \langle w, \boldsymbol{\mu} \rangle - \mu_i + 2\sqrt{2K\widehat{V}_t(X_i, \langle w, \boldsymbol{X} \rangle)}\,\alpha(t,\delta) \\
&\quad + 14K\|w - e_i\|_1 \alpha^2(t,\delta) \\
&\leq \hat{\Gamma}_i(w, t, \delta) + \sqrt{2K\mathrm{Var}(X_i - \langle w, \boldsymbol{X} \rangle)}\alpha(t,\delta) + 2\sqrt{2K\widehat{V}_t(X_i - \langle w, \boldsymbol{X} \rangle)}\alpha(t,\delta) \\
&\quad + \frac{42K}{3}\|w - e_i\|_1 \alpha^2(t,\delta) \\
&\leq \hat{\Gamma}_i(w, t, \delta) + 3\sqrt{2K\mathrm{Var}(X_i - \langle w, \boldsymbol{X} \rangle)}\alpha(t,\delta) + 9K\alpha^2(t,\delta) + \frac{42K}{3}\|w - e_i\|_1 \alpha^2(t,\delta) \\
&\leq 3\sqrt{2K\mathrm{Var}(X_i - \langle w, \boldsymbol{X} \rangle)}\alpha(t,\delta) + 25K\|w - e_i\|_1 \alpha^2(t,\delta),
\end{aligned}
$$

where we used (13a) in the second line and (13b) with $\hat{\Gamma}_i(w, t, \delta_t) \leq 0$ in the third line. Suppose that $\langle w, \boldsymbol{\mu} \rangle > \mu_i$. Solving the inequality above in $\alpha(t, \delta)$, gives

$$
\begin{aligned}
\alpha(t,\delta) &\geq \frac{\sqrt{18K\mathrm{Var}(X_i - \langle w, \boldsymbol{X} \rangle) + 100K\|w - e_i\|_1(\langle w, \boldsymbol{\mu} \rangle - \mu_i)} - 3\sqrt{2K\mathrm{Var}(X_i - \langle w, \boldsymbol{X} \rangle)}}{50K\|w - e_i\|_1} \\
&= \frac{2(\langle w, \boldsymbol{\mu} \rangle - \mu_i)}{\sqrt{18K\mathrm{Var}(X_i - \langle w, \boldsymbol{X} \rangle) + 100K\|w - e_i\|_1(\langle w, \boldsymbol{\mu} \rangle - \mu_i)} + 3\sqrt{2K\mathrm{Var}(X_i - \langle w, \boldsymbol{X} \rangle)}}.
\end{aligned}
$$

Therefore, we have

$$t \leq K \log(\delta_t^{-1}) \left( \frac{18\text{Var}(X_i - \langle w, \boldsymbol{X} \rangle)}{(\langle w, \boldsymbol{\mu} \rangle - \mu_i)^2} + \frac{50\|w - e_i\|_1}{\langle w, \boldsymbol{\mu} \rangle - \mu_i} \right)$$

$$\leq 36K \log(\delta_t^{-1}) \Xi_i(w).$$

If $\langle w, \boldsymbol{\mu} \rangle \leq \mu_i$, then $\Xi_i(w) = +\infty$ and the inequality above is straightforward.

$\square$

**Lemma E.6.** *Let $i, j$ and $k \in [\![K]\!]$, we have:*

$$\Lambda'_{ij} \leq \max\{\Lambda'_{ik}, \Lambda'_{kj}\}$$

$$\Lambda_{ij} \leq \max\{\Lambda_{ik}, \Lambda_{kj}\}$$

*Proof.* Let $i, j$ and $k \in [\![K]\!]$. Suppose that $\mu_j \leq \mu_i$. Hence, for any $k \in [\![K]\!]$, $\mu_k \leq \mu_i$ or $\mu_k \geq \mu_j$. Therefore

$$\max\{\Lambda'_{ik}; \Lambda'_{kj}\} = +\infty,$$

which proves the result. The same argument applies to the quantities $\Lambda_{ik}$ and $\Lambda_{kj}$.

Now suppose that $\mu_j > \mu_i$ (hence $\Lambda'_{ij} < +\infty$ and $\Lambda_{ij} < +\infty$). Let us start by proving the first claim. Recall the definition in Section A:

$$\Lambda'_{ij} := \frac{V_{ij}}{(\mu_j - \mu_i)^2}.$$

We have

$$\frac{\sqrt{V_{ij}}}{\mu_j - \mu_i} \leq \frac{\sqrt{V_{ik}} + \sqrt{V_{kj}}}{(\mu_j - \mu_k) + (\mu_k - \mu_i)}$$

$$\leq \max\left\{ \frac{\sqrt{V_{kj}}}{\mu_j - \mu_k}, \frac{\sqrt{V_{ik}}}{\mu_k - \mu_i} \right\},$$

where the first line follows by the triangle inequality and the second is a consequence of the inequality $\frac{a+b}{c+d} \leq \max\{\frac{a}{c}, \frac{b}{d}\}$ (Lemma K.3), which proves the first claim.

Moreover, we have using the result above: for any $k$ such that $\mu_i < \mu_k < \mu_j$

$$\Lambda_{ij} = \frac{V_{ij}}{(\mu_j - \mu_i)^2} + \frac{3}{\mu_j - \mu_i}$$

$$\leq \max\left\{ \frac{V_{ik}}{(\mu_k - \mu_i)^2} + \frac{3}{\mu_j - \mu_i}, \frac{V_{kj}}{(\mu_j - \mu_k)^2} + \frac{3}{\mu_j - \mu_i} \right\}$$

$$\leq \max\left\{ \frac{V_{ik}}{(\mu_k - \mu_i)^2} + \frac{3}{\mu_k - \mu_i}, \frac{V_{kj}}{(\mu_j - \mu_k)^2} + \frac{3}{\mu_j - \mu_k} \right\}$$

$$= \max\{\Lambda_{ik}, \Lambda_{kj}\}.$$

If $k$ is such that $\mu_k < \mu_i$ or $\mu_k > \mu_j$, we have $\max\{\Lambda_{ik}, \Lambda_{kj}\} = +\infty$, which proves the statement.

$\square$

## F  Proof of Theorem 4.1

The proof of Theorem 4.1 follows the same steps as the proof of Theorem 5.1.

For any $i \in [\![K]\!] \setminus \{i^*\}$, let us define $\Upsilon_i$ by

$$\Upsilon_i := \underset{j \in [\![K]\!]}{\text{Arg Min}} \, \Lambda_{ij}. \tag{32}$$

**Lemma F.1.** *Consider Algorithm 2 with inputs $\delta \in (0, 1)$. If $(\mathcal{B}_1)$ defined in (13) holds, then for any $i \in [\![K]\!] \setminus \{i^*\}$ and $t \geq 1$:*

*If $i \in S_t$, then $\Upsilon_i \cap C_t \neq \emptyset$, where $\Upsilon_i$ is defined in (32).*

*Proof.* Suppose that $(\mathcal{B}_1)$ holds. Let $t \geq 1$, $i \in [\![K]\!] \setminus \{i^*\}$. Proceeding by proof via contradiction, suppose that $\Upsilon_i \cap C_t = \emptyset$. This implies in particular that all elements in $\Upsilon_i$ were eliminated prior to $t$. Let $j$ denote the element of $\Upsilon_i$ with the largest mean:

$$j \in \underset{l \in \Upsilon_i}{\mathrm{Arg\,Max}}\{\mu_l\}.$$

Let $s$ denote the round where $j$ has failed the test (i.e. $\exists k \in C_s$, $\hat{\Delta}_{jk}(s, \delta) > 0$).

Hence, using Lemma E.3, we have

$$\frac{1}{4} \log(\delta_s^{-1}) \Lambda_{jk} \leq s. \tag{33}$$

Moreover, $j$ was kept for testing up to round $82\,s$ (i.e. $j \in C_{82s}$) and $82s < t$ (since $j \notin C_t$). At round $82s$ we necessarily had $\hat{\Delta}_{ij}(82s, \delta) \leq 0$.

Therefore, using Lemma E.3

$$82s \leq 25 \log(\delta_{82s}^{-1}) \Lambda_{ij}. \tag{34}$$

Combining (33) and (34) gives

$$\frac{82}{4} \log(\delta_s^{-1}) \Lambda_{jk} \leq 25 \log(\delta_{82s}^{-1}) \Lambda_{ij}.$$

Therefore

$$\Lambda_{jk} \leq \frac{50}{82}\left(1 + \frac{\log(82)}{\log(1/\delta_s)}\right)\Lambda_{ij} \leq \Lambda_{ij}, \tag{35}$$

recall that $\delta < 1/4$, $s \geq 3$, $K \geq 3$, therefore: $\frac{50}{82}\left(1 + \frac{\log(82)}{\log(1/\delta_s)}\right) \leq 1$. Using Lemma E.6, we have

$$\Lambda_{ik} \leq \max\{\Lambda_{ij}, \Lambda_{jk}\}. \tag{36}$$

We plug the bound $\Lambda_{jk} \leq \Lambda_{ij}$ from (35) into (36) and obtain $\Lambda_{ik} \leq \Lambda_{ij}$. Therefore $k \in \Upsilon_i$.

To conclude, recall that $k$ eliminates $j$, hence $\mu_k > \mu_j$. The contradiction arises from $k \in \Upsilon_i$ and the definition of $j$ as the element with largest mean in $\Upsilon_i$. □

We introduce the following notation. For $i \in [\![K]\!]$ and $t \geq 1$ let $N_{i,t}$ denote the number of queries made for arm $i$ up to round $t$

$$N_{i,t} := \sum_{s=1}^{t} \mathbb{1}(i \in C_s). \tag{37}$$

**Lemma F.2.** *Consider Algorithm 2 with inputs $\delta \in (0, 1)$. If $(\mathcal{B}_1)$ defined in (13) holds, then we have for each $i \in [\![K]\!] \setminus \{i^*\}$:*

$$\forall t \geq 1: \quad N_{i,t} \leq 1368 \min_{j \in [\![K]\!]}\{\Lambda_{ij}\} \log(216K^2\delta^{-1} \min_{j \in [\![K]\!]}\{\Lambda_{ij}\}),$$

*where $N_{i,t}$ is defined in (37).*

*Proof.* Suppose $(\mathcal{B}_1)$ holds. Let $i \in [\![K]\!] \setminus \{i^*\}$ and $t \geq 1$. Let $u$ denote the last round such that $i \in S_u$. Lemma F.1 states that $\Upsilon_i \cap C_u \neq \emptyset$, where $\Upsilon_i$ is defined in (45). Let $j \in \Upsilon_i \cap C_u$, since $i \in S_u$, we necessarily have

$$\hat{\Delta}_{ij}(u - 1, \delta) \leq 0.$$

Using Lemma E.3, we have

$$u - 1 \leq \frac{25}{2} \log(\delta_{u-1}^{-1}) \Lambda_{ij}.$$

Recall that $u$ is the last round such that $i \in S_u$, hence $i \notin C_{19u+1}$. Therefore, for any $t \geq 1$

$$N_{i,t} = 82u \leq 1013 \log(\delta_u^{-1})\Lambda_{ij}$$
$$\leq 2026 \min_{j \in [\![K]\!]}\{\Lambda_{ij}\} \log(216K^2\delta^{-1} \min_{j \in [\![K]\!]}\{\Lambda_{ij}\}),$$

where we used Lemma K.4 with $x = u$ and $c = \delta/(2K^2)$. □

**Proof for Theorem 4.1.** Suppose Assumptions 1 and $2 - \mathcal{B}$ hold. Consider Algorithm 2 with input $\delta \in (0, 1/4)$. Suppose that event $(\mathcal{B}_1)$ holds.

We have by definition of the total number of queries made $N$:

$$N = \sum_{i \in [\![K]\!] \setminus \{i^*\}} N_{i,t}.$$

Therefore, Lemma F.2 gives the result. For the second result, consider Algorithm 2 without line (12) (we stop sampling arms directly after their elimination from $S_t$). Lemma E.1 guarantees with probability at least $1 - \delta$, the optimal arm $i^*$ always belongs to $S_t$. Therefore, Lemma E.5 guarantees that after at most $\frac{25}{2} \log(\delta_t^{-1}) \Lambda_{ii^*}$ rounds, we have $\hat{\Delta}_{ii^*}(t, \delta) > 0$, which leads to the elimination of $i$.

## G   Proof of Theorem 5.1

For any $i \in [\![K]\!] \setminus \{i^*\}$, let us define $\Upsilon_i'$ by

$$\Upsilon_i' := \underset{j \in [\![K]\!]}{\mathrm{Arg\,Min}} \left\{ \Lambda_{ij}' \vee \frac{1}{4} \right\}. \tag{38}$$

**Lemma G.1.** *Consider Algorithm 2 with inputs $\delta \in (0, 1)$. If $(\mathcal{A}_1)$ defined in (13) holds, then for any $i \in [\![K]\!] \setminus \{i^*\}$ and $t \geq 1$:*

*If $i \in S_t$, then $\Upsilon_i' \cap C_t \neq \emptyset$, where $\Upsilon_i'$ is defined in (38).*

*Proof.* Suppose that $(\mathcal{A}_1)$ holds. Let $t \geq 2$, $i \in [\![K]\!] \setminus \{i^*\}$. Proceeding by proof via contradiction, suppose that $\Upsilon_i' \cap C_t = \emptyset$. This implies in particular that all elements in $\Upsilon_i'$ were eliminated prior to $t$. Let $j$ denote the element of $\Upsilon_i'$ with the largest mean:

$$j \in \underset{l \in \Upsilon_i'}{\mathrm{Arg\,Max}} \{\mu_l\}.$$

Let $s$ denote the round where $j$ has failed the test (i.e. $\exists k \in C_s, \hat{\Delta}_{jk}'(s, \delta) > 0$).

Hence, using Lemma E.4, we have

$$\frac{1}{2} \Lambda_{jk}' \log(1/\delta_s) \leq s - 1. \tag{39}$$

We consider two cases:

**Case 1:** $\Lambda_{ij}' \geq \frac{1}{5}$.   Observe that $j$ was kept for testing up to round $35s$ (i.e. $j \in C_{82s}$) and $82s < t$ (since $j \notin C_t$). At round $82s$ we necessarily had $\hat{\Delta}_{ij}'(82s, \delta) \leq 0$.

Therefore, using Lemma E.4

$$82s - 1 \leq 25\Lambda_{ij}' \log(1/\delta_{35s}). \tag{40}$$

Combining (39) and (40) gives

$$\frac{82}{2} \Lambda_{jk}' \log(1/\delta_s) \leq 25\Lambda_{ij}' \log(1/\delta_{82s}).$$

Therefore

$$\Lambda_{jk}' \leq \frac{50}{82}\left(1 + \frac{\log(82)}{\log(\delta_s^{-1})}\right)\Lambda_{ij}' \leq \Lambda_{ij}'. \tag{41}$$

Using Lemma E.6, we have

$$\Lambda_{ik}' \leq \max\{\Lambda_{ij}', \Lambda_{jk}'\}. \tag{42}$$

We plug the bound $\Lambda_{jk}' \leq \Lambda_{ij}'$ from (41) into (42) and obtain $\Lambda_{ik}' \leq \Lambda_{ij}'$. Therefore $k \in \Upsilon_i'$.

To conclude, recall that $k$ eliminates $j$, hence $\mu_k > \mu_j$. The contradiction arises from $k \in \Upsilon_i'$ and the definition of $j$ as the element with largest mean in $\Upsilon_i'$.

**Case 2: $\Lambda'_{ij} < \frac{1}{5}$.** As in the previous case, observe that $j$ was kept for testing up to round $82s$ (i.e. $j \in C_{82s}$) and $82s < t$ (since $j \notin C_t$). At round $82s$ we necessarily had $\hat{\Delta}_{ij}(82s, \delta) \leq 0$.

Therefore, using Lemma E.4

$$82s - 1 \leq \frac{3}{2 \log\big(1/(3\sqrt{2}\Lambda'_{ij})\big)} \log(1/\delta_{82s}). \tag{43}$$

Combining (39) and (43) gives

$$\frac{82}{2} \Lambda'_{jk} \log(1/\delta_s) \leq \frac{3 \log(1/\delta_{82s})}{2 \log(\sqrt{2}/(5\Lambda'_{ij}))}.$$

Therefore

$$\begin{aligned}
\Lambda'_{jk} &\leq \frac{3}{82} \left( 1 + \frac{\log(82)}{\log(1/\delta_s)} \right) \frac{1}{\log(\sqrt{2}/(5\Lambda'_{ij}))} \\
&\leq \frac{6}{82 \log(1/(3\sqrt{2}\Lambda'_{ij}))}.
\end{aligned}$$

Recall that using Lemma E.6, we have: $\Lambda'_{ik} \leq \max\{\Lambda'_{ij}, \Lambda'_{jk}\}$. This implies necessarily that $\Lambda'_{ik} \leq \Lambda'_{jk}$. Otherwise if the maximum of the l.h.s is $\Lambda'_{ij}$, we get $\Lambda'_{ij} \leq \Lambda'_{ij}$ then $k \in \Upsilon'_i$, therefore by definition of $j$ as the largest mean of $\Upsilon'_i$: $\mu_k \leq \mu_j$, which contradicts the fact that $k$ eliminated $j$. We conclude that (using $\Lambda'_{ij} < 1/5$):

$$\Lambda'_{ik} \leq \frac{6}{82 \log(\sqrt{2}/(5\Lambda'_{ij}))} < \frac{1}{4}.$$

Therefore $k \in \operatorname{Arg\,Min}_{j \in \llbracket K \rrbracket}\{\Lambda'_{ij} \vee \frac{1}{4}\}$, which similarly to the case above, leads to a contradiction with the definition of $j$.

$\square$

We introduce the following notation. For $i \in \llbracket K \rrbracket$ and $t \geq 1$ let $N_{i,t}$ denote the number of queries made for arm $i$ up to round $t$

$$N_{i,t} := \sum_{s=1}^{t} \mathbb{1}(i \in C_s). \tag{44}$$

**Lemma G.2.** *Consider Algorithm 2 with inputs $\delta \in (0, 1)$. If $(\mathcal{A}_1)$ defined in (22) holds, then we have for each $i \in \llbracket K \rrbracket \setminus \{i^*\}$:*

$$\forall t \geq 1: \quad N_{i,t} \leq 4100 \log(216 K \delta^{-1} \min_{j \in \llbracket K \rrbracket}\{\Lambda'_{ij} \vee 1\})) \min_{j \in \llbracket K \rrbracket}\left\{\Lambda'_{ij} \vee \frac{1}{4}\right\},$$

*where $N_{i,t}$ is defined in (37).*

*Proof.* Suppose $(\mathcal{A}_1)$ holds. Let $i \in \llbracket K \rrbracket \setminus \{i^*\}$ and $t \geq 1$. Let $u$ denote the last round such that $i \in S_u$. Lemma G.1 states that $\Upsilon'_i \cap C_u \neq \emptyset$, where $\Upsilon'_i$ is defined in (45). Let $j \in \Upsilon'_i \cap C_u$, since $i \in S_u$, we necessarily have

$$\hat{\Delta}'_{ij}(u - 1, \delta) \leq 0.$$

Using Lemma E.4, we have

$$u - 1 \leq \frac{25}{\log\big(1 + 1/\Lambda'_{ij}\big)} \log(\delta_{u-1}^{-1}).$$

Recall that $u$ is the last round such that $i \in S_u$, hence $i \notin C_{35u+1}$. Therefore, for any $t \geq 1$

$$
\begin{aligned}
N_{i,t} = 82u &\leq \frac{2050}{\log\big(1 + 1/\Lambda'_{ij}\big)} \log(\delta_u^{-1}) \\
&\leq 4100 \log(216K(\Lambda'_{ij} + 1)\delta^{-1}) \frac{1}{\log\big(1 + 1/\Lambda'_{ij}\big)} \\
&\leq 4100 \log(216K(\Lambda'_{ij} + 1)\delta^{-1}) \frac{1}{\log\big(1 + 1/\Lambda'_{ij}\big)} \\
&\leq 4100 \log(216K(\Lambda'_{ij} + 1)\delta^{-1}) \frac{1}{\log\big(1 + 1/(\Lambda'_{ij} \vee 0.25)\big)} \\
&\leq 4100 \log(216K(\Lambda'_{ij} + 1)\delta^{-1}) \frac{8}{7}(\Lambda'_{ij} \vee 0.25) \\
&\leq 4100 \log(216K(\Lambda'_{ij} + 1)\delta^{-1}) \min_{j \in [\![K]\!]} (\Lambda'_{ij} \vee 0.25),
\end{aligned}
$$

$j \in \Upsilon_i$ and Lemma K.4 with $x = u$ and $c = \delta/(2K^2)$. $\qquad\square$

**Proof for Theorem 5.1** Suppose Assumptions 1 and 2 $-\mathcal{G}$. Consider Algorithm 2 with input $\delta \in (0, 1/4)$. Suppose that event $(\mathcal{B}_1)$ holds.

We have by definition of the total number of queries made $N$:

$$
N = \sum_{i \in [\![K]\!] \setminus \{i^*\}} N_{i,t}.
$$

Therefore, Lemma F.2 gives the result. For the second result, consider Algorithm 2 without line (12) (we stop sampling arms directly after their elimination from $S_t$). Lemma E.1 guarantees with probability at least $1 - \delta$, the optimal arm $i^*$ always belongs to $S_t$. Therefore, Lemma E.5 guarantees that after at most $\frac{25}{2} \log(\delta_t^{-1}) \frac{1}{\log\big(1 + 1/\Lambda'_{ii^*}\big)}$ rounds, we have $\hat{\Delta}'_{ii^*}(t, \delta) > 0$, which leads to the elimination of $i$.

## H  Proof of Theorem B.1

We provide the same type of guarantees for Algorithm 3.

For any $u, v \in \boldsymbol{B}_1^+$, we overload the notation $\Xi_i(u)$ into

$$
\Xi_u(v) := \begin{cases} +\infty & \text{if } \langle u, \boldsymbol{\mu}\rangle \leq \langle v, \boldsymbol{\mu}\rangle \\ \max\left\{ \frac{\mathrm{Var}(\langle \boldsymbol{X}, u\rangle - \langle \boldsymbol{X}, v\rangle)}{(\langle u, \boldsymbol{\mu}\rangle - \langle v, \boldsymbol{\mu}\rangle)^2}, \frac{3\|u - v\|_1}{\langle v, \boldsymbol{\mu}\rangle - \langle u, \boldsymbol{\mu}\rangle} \right\} & \text{otherwise} \end{cases}
$$

In particular we have $\Xi_{e_i}(w) = \Xi_i(w)$, where $(e_i)_{i \in [\![K]\!]}$ is the canonical basis of $\mathbb{R}^K$. We say that an arm $i \in [\![K]\!]$ has failed the $\Gamma$-test at round $t$, if

$$
\sup_{w \in \boldsymbol{B}_1^+(C_t \setminus \{i\})} \hat{\Gamma}_i(w, t, \delta) > 0.
$$

**Lemma H.1.** *Let $i \in [\![K]\!]$, $u, v \in \boldsymbol{B}_1^+([\![K]\!] \setminus \{i\})$, we have*

$$
\Xi_i(v) \leq \max\{\Xi_i(u), \Xi_u(v)\}.
$$

*Proof.* Let $i \in [\![K]\!]$ and $u, v \in \boldsymbol{B}_1^+([\![K]\!] \setminus \{i\})$. Suppose that $\mu_i < \langle u, \boldsymbol{\mu}\rangle < \langle v, \boldsymbol{\mu}\rangle$. We have

$$
\begin{aligned}
\frac{\sqrt{\mathrm{Var}(X_i - \langle v, \boldsymbol{X}\rangle)}}{\langle v, \boldsymbol{\mu}\rangle - \mu_i} &\leq \frac{\sqrt{\mathrm{Var}(X_i - \langle u, \boldsymbol{X}\rangle)} + \sqrt{\mathrm{Var}(\langle u, \boldsymbol{X}\rangle - \langle v, \boldsymbol{X}\rangle)}}{(\langle v, \boldsymbol{\mu}\rangle - \langle u, \boldsymbol{\mu}\rangle) + (\langle u, \boldsymbol{\mu}\rangle - \mu_i)} \\
&\leq \max\left\{ \frac{\sqrt{\mathrm{Var}(X_i - \langle u, \boldsymbol{X}\rangle)}}{\langle v, \boldsymbol{\mu}\rangle - \langle u, \boldsymbol{\mu}\rangle}, \frac{\sqrt{\mathrm{Var}(\langle u, \boldsymbol{X}\rangle - \langle v, \boldsymbol{X}\rangle)}}{\langle u, \boldsymbol{\mu}\rangle - \mu_i} \right\},
\end{aligned}
$$

where the first line follows by the triangle inequality and the second is a consequence of the inequality $\frac{a+b}{c+d} \leq \max\{\frac{a}{c}, \frac{b}{d}\}$ for positive numbers (Lemma K.3). Moreover we have

$$
\begin{aligned}
\frac{\|v - e_i\|_1}{\langle v - e_i, \boldsymbol{\mu}\rangle} &\leq \frac{\|v - u\|_1 + \|u - e_i\|_1}{\langle v - u, \boldsymbol{\mu}\rangle + \langle u - e_i, \boldsymbol{\mu}\rangle} \\
&\leq \max\left\{ \frac{\|v - u\|_1}{\langle v - u, \boldsymbol{\mu}\rangle}, \frac{\|u - e_i\|_1}{\langle u - e_i, \boldsymbol{\mu}\rangle} \right\},
\end{aligned}
$$

where we used the inequality $\frac{a+b}{c+d} \leq \max\{\frac{a}{c}, \frac{b}{d}\}$ for positive numbers (Lemma K.3).

Combining the previous bounds, we obtain the result.

If $\mu_i \geq \langle u, \boldsymbol{\mu}\rangle$ or $\langle u, \boldsymbol{\mu}\rangle \geq \langle v, \boldsymbol{\mu}\rangle$. We have

$$
\max\{\Xi_i(u); \Xi_u(v)\} = +\infty,
$$

which proves the result. $\qquad\square$

For any $i \in [\![K]\!] \setminus \{i^*\}$, let us define $\Psi_i$ by

$$
\Psi_i := \underset{w \in \boldsymbol{B}_1^+([\![K]\!]\setminus\{i\})}{\operatorname{Arg\,Min}} \Xi_i(w). \tag{45}
$$

**Lemma H.2.** *Consider Algorithm 3 with input $\delta \in (0,1)$. If $(\mathcal{B}_2)$ defined in (13) holds, then for any $i \in [\![K]\!] \setminus \{i^*\}$, $t \geq 1$:*

*If $i \in S_t$, then there exists a vector $w^* \in \Psi_i$ such that: $\operatorname{supp}(w^*) \subseteq C_t$.*

*Proof.* Let $t \geq 1$, $i \in [\![K]\!] \setminus \{i^*\}$. We take $w^*$ to be one of the vectors from the set $\Psi_i$, such that its support was jointly queried the most up to round $t$. More formally:

$$
w^* \in \underset{w \in \Psi_i}{\operatorname{Arg\,Max}}\{\langle w, \boldsymbol{\mu}\rangle\}.
$$

Proceeding by proof via contradiction, we suppose that $\operatorname{supp}(w^*) \not\subset C_t$. Then, we will build a vector $w' \in \boldsymbol{B}_1^+$, such that $\langle w^*, \boldsymbol{\mu}\rangle < \langle w', \boldsymbol{\mu}\rangle$, the contradiction follows from the definition of $w^*$. Let $j$ be the first eliminated element in $\operatorname{supp}(w^*)$. Let $s$ denote the round where $j$ has failed the $\Gamma$-test (i.e. $\exists \tilde{w} \in \boldsymbol{B}_1^+, \hat{\Gamma}_j(\tilde{w}, s, \delta) > 0$).

Let us define $w' \in \mathbb{R}^K$ as follows: $w_j' = w_j^* \tilde{w}_j$ and for $k \in [\![K]\!] \setminus \{j\}$, $w_k' = w_k^* + w_j^* \tilde{w}_k$. Recall that

$$
\begin{aligned}
\|w'\|_1 &= \sum_{k \in [\![K]\!]\setminus\{j\}} w_k^* + \sum_{k \in [\![K]\!]} w_j^* \tilde{w}_k \\
&= 1 - w_j^* + w_j^* \|\tilde{w}\|_1 \\
&= 1,
\end{aligned}
$$

where we used the fact that $\|\tilde{w}\|_1 = 1$. We conclude that $w' \in \boldsymbol{B}_1^+$.

Let us show that $w' \in \Psi_i$. We have

$$
\begin{aligned}
\langle w^* - w', \boldsymbol{\mu}\rangle &= w_j^*(1 - \tilde{w}_j)\mu_j + \sum_{k \in [\![K]\!]\setminus\{j\}} (w_k^* - w_k^* - w_j^* \tilde{w}_k)\mu_k \\
&= w_j^* \mu_j - w_j^* \tilde{w}_j \mu_j - w_j^* \sum_{k \in [\![K]\!]\setminus\{j\}} \tilde{w}_k u_k \\
&= w_j^*(\mu_j - \langle \tilde{w}, \boldsymbol{\mu}\rangle). \tag{46}
\end{aligned}
$$

Moreover

$$
\begin{aligned}
\|w^* - w'\|_1 &= \sum_{k=1}^{K} |w_k^* - w_k'| \\
&= w_j^* |1 - \tilde{w}_j| + w_j^* \sum_{k \in [\![K]\!]\setminus\{j\}} \tilde{w}_k \\
&= w_j^* \|\tilde{w} - e_j\|_1.
\end{aligned}
$$

Using (46) we have

$$\Xi_{w^*}(w') = \max\left\{\frac{\mathrm{Var}(\langle w^* - w', \boldsymbol{X}\rangle)}{(\langle w^* - w', \boldsymbol{\mu}\rangle)^2}; \frac{\|w^* - w'\|_1}{\langle w' - w^*, \boldsymbol{\mu}\rangle}\right\}$$

$$= \max\left\{\frac{\mathrm{Var}(w_j(X_j - \langle \tilde{w}, \boldsymbol{X}\rangle))}{w_j^2(\mu_j - \langle \tilde{w}, \boldsymbol{u}\rangle)^2}; \frac{\|\tilde{w} - e_j\|_1}{\langle \tilde{w}, \boldsymbol{\mu}\rangle - \mu_j}\right\}$$

$$= \Xi_j(\tilde{w}).$$

Therefore, using Lemma H.1

$$\Xi_i(w') \leq \max\{\Xi_i(w^*); \Xi_{w^*}(w')\}$$
$$= \max\{\Xi_i(w^*); \Xi_j(\tilde{w})\}. \tag{47}$$

Recall that $\hat{\Gamma}_j(\tilde{w}, s, \delta) > 0$. Hence using Lemma E.5, we have

$$K \log(2K^2\delta_s^{-1})\Xi_j(\tilde{w}) \leq s. \tag{48}$$

Moreover, since $j$ failed the $\Gamma$-test at round $s$, we have by construction of Algorithm 3 $j \in C_{98s}$. Recall that $j$ is the first element of the support of $w^*$ that was eliminated, then we necessarily have $\mathrm{supp}(w^*) \subset C_{98s}$. Since we assumed that $\mathrm{supp}(w^*) \not\subset C_t$, we have $98s < t$, hence $i \in C_{98s}$ and $\hat{\Gamma}_i(w^*, 98s, \delta) \leq 0$. Using Lemma E.5

$$98s \leq 36K \log(2K^2\delta_{98s}^{-1})\, \Xi_i(w^*). \tag{49}$$

Combining inequalities (48) and (49), we have

$$98K \log(2K^2\delta_s^{-1})\Xi_j(\tilde{w}) < 36K \log(2K^2\delta_{98s}^{-1})\Xi_i(w^*).$$

Therefore

$$\Xi_j(\tilde{w}) \leq \frac{36}{98}\frac{\log(2K^2\delta_{98s}^{-1})}{\log(2K^2\delta_s^{-1})}\Xi_i(w^*)$$

$$\leq \frac{36}{98}\left(1 + \frac{\log(98)}{\log(\delta_s^{-1})}\right)\Xi_i(w^*)$$

$$\leq \Xi_i(w^*).$$

Combining the bound above with (47), we conclude that $\Xi_i(w') \leq \Xi_i(w^*)$. Hence $w' \in \Psi_i$.

Finally, recall that by (46) $\langle w', \boldsymbol{\mu}\rangle > \langle w^*, \boldsymbol{\mu}\rangle$ (since $\tilde{w}$ eliminated $j$: $\langle \tilde{w}, \mu_j\rangle - \mu_j > 0$). The conclusion follows from $w' \in \Psi_i$ and the definition of $w^*$. $\qquad\square$

**Lemma H.3.** *Consider Algorithm 3 with input $\delta \in (0, 1)$. If $(\mathcal{B}_2)$ defined in (13) holds, then we have for each $i \in [\![K]\!]$, $t \geq 1$:*

$$N_{i,t} \leq 10540 \log\big(1296K^2\Xi_i(w^*)\delta^{-1}\big)K\Xi_i(w^*).$$

*Proof.* Suppose $(\mathcal{A}_2)$ holds. Let $i \in [\![K]\!] \setminus \{i^*\}$ and $t \geq 1$. Let $u$ denote the last round such that $i \in S_u$. Lemma H.2 states that there exists $w^* \in \Psi_i$ such that $\mathrm{supp}(w^*) \subset C_u$, where $\Psi_i$ is defined in (32). Since $i \in S_u$, we necessarily have:

$$\hat{\Gamma}_i(w^*, u - 1, \delta) \leq 0.$$

Using Lemma E.3, we have

$$u - 1 \leq 108K \log(2K^2\delta_{u-1}^{-1})\Xi_i(w^*).$$

Recall that $u$ is the last round such that $i \in S_u$, therefore $i \notin C_{98u+1}$. Hence, for any $t \geq 1$

$$N_{i,t} = 98u \leq 10540K \log(2K^2\delta_u^{-1})\Xi_i(w^*)$$

$$\leq 10540 \log\big(1296K^2\Xi_i(w^*)\delta^{-1}\big)K\Xi_i(w^*),$$

where we used Lemma K.4 with $x = u$ and $c = \delta/(2K^2)$. $\qquad\square$

The conclusion for the proof of Theorem B.1 is similar to the conclusion of the proof of Theorem 4.1.

# I  Proof of Theorem 6.1

Without loss of generality assume that $\mu_1 > \mu_2 \geq \cdots \geq \mu_K$. For any Bernoulli variables $(X_i)_i$ with means $(\mu_i)_i$. Fix a sequence of positive numbers $(V_{i1})_{i\geq 2}$ such that for each $i \geq 2$:

$$(\mu_1 - \mu_i) - (\mu_1 - \mu_i)^2 \leq \text{Var}(X_1 - X_i) \leq \min(2 - (\mu_1 + \mu_i); \; \mu_1 + \mu_i) - (\mu_1 - \mu_i)^2. \quad (50)$$

Below we build $K$ Bernoulli variables $(X_i)_{i\in\llbracket K\rrbracket}$ such that $\mathbb{E}[X_i] = \mu_i$, $\text{Var}(X_i - X_1) \leq V_{i1}$ and $1/4 \leq \mu_i \leq 3/4$, for all $i \in \llbracket K\rrbracket$. Recall that the set of Bernoulli variables satifying the last constraints is denoted $\mathbb{B}_K(\boldsymbol{\mu}, \boldsymbol{V})$.

**Building a distribution in $\mathbb{B}_K(\boldsymbol{\mu}, \boldsymbol{V})$:**  Let $(U_t)_{t\geq 1}$ and $(W_{i,t})_{i\in\llbracket K\rrbracket, t\geq 1}$ denote sequences of independent variables following the uniform distribution on the interval $[0, 1]$. Let $(a_i, b_i)_{i\in\llbracket K\rrbracket}$ denote a sequence of numbers in $[0, 1]$ to be specified later. We define the arms variables $(X_i)_{i\in\llbracket K\rrbracket}$ as follows:

- $X_{1,t} = \mathbb{1}(U_t \leq \mu_1)$ for each $t \geq 1$.
- For $i \geq 2$, $t \geq 1$. We consider two cases:
    - If $V_{i1} \leq \mu_1 + \mu_i - \mu_1^2 - \mu_i^2$: Let $X_{i,t} = \mathbb{1}(\{U_t \leq a_i\} \text{ or } \{W_{i,t} \leq b_i\})$.
    - If $V_{i1} \geq \mu_1 + \mu_i - \mu_1^2 - \mu_i^2$: Let $X_{i,t} = \mathbb{1}(W_{i,t} \leq \mu_i)$.

Now let us specify our choice for the sequences $(a_i)$ and $(b_i)$.

**If $V_{i1} \leq \mu_1 + \mu_i - \mu_1^2 - \mu_i^2$:**  The first constraint is with respect to the means of $(X_i)$. We need to have for each $i \in \llbracket K\rrbracket$: $\mathbb{E}[X_i] = \mu_i$, this implies the following for each $i \geq 2$:

$$\mathbb{E}[X_i] = \mathbb{E}[\mathbb{1}(\{U_t \leq a_i\} \text{ or } \{W_{i,t} \leq b_i\})]$$
$$= a_i + b_i - a_i b_i.$$

We therefore have

$$a_i + b_i - a_i b_i = \mu_i \quad (51)$$

The second constraint is with tespect to the variance of the variable $(X_1 - X_i)$, we set $\text{Var}(X_1 - X_i) = V_{i1}$. This implies the following

$$\text{Var}(X_1 - X_i) = \mathbb{E}\left[(\mathbb{1}(U_t \leq \mu_1) - \mathbb{1}(\{U_t \leq a_i\} \text{ or } \{W_{i,t} \leq b_i\}))^2\right] - (\mu_1 - \mu_i)^2$$
$$= \mu_1 + \mu_i - 2(a_i + (\mu_1 - a_i)b_i) - (\mu_1 - \mu_i)^2.$$

We therefore have that $a_i$ and $b_i$ satisfy

$$\mu_1 + \mu_i - 2(a_i + (\mu_1 - a_i)b_i) - (\mu_1 - \mu_i)^2 = V_{i1}. \quad (52)$$

Solving in $a_i$ and $b_i$ for the system: (51) and (52) is:

$$b_i = \frac{V_{i1} - \left((\mu_1 - \mu_i) - (\mu_1 - \mu_i)^2\right)}{2(1 - \mu_1)}$$
$$a_i = \frac{\mu_i - b_i}{1 - b_i}$$

Observe that when $V_{i1} \leq \mu_1 + \mu_i - \mu_1^2 - \mu_i^2$, we have :

$$b_i = \frac{V_{i1} - (\mu_1 - \mu_i) + (\mu_1 - \mu_i)^2}{2(1 - \mu_1)}$$
$$\leq \frac{2\mu_i - 2\mu_i\mu_1}{2(1 - \mu_1)}$$
$$= \mu_i.$$

Therefore $b_i \leq \mu_i$. Moreover, using Lemma K.8, we have $b_i \geq 0$. We conclude that $b_i \in [0, \mu_i]$, which implies that $a_i \in [0, 1]$.

**If $V_{i1} \geq \mu_1 + \mu_i - \mu_1^2 - \mu_i^2$:** Let us prove in case that we have $\text{Var}(X_1 - X_i) \leq V_{i1}$. Recall that $X_i$ and $X_1$ are independent. Therefore:

$$\begin{aligned} \text{Var}(X_1 - X_i) &= \mu_1 + \mu_i - 2\mu_1\mu_i - (\mu_1 - \mu_i)^2 \\ &\leq V_{i1} + \mu_1^2 + \mu_i^2 - 2\mu_i\mu_1 - (\mu_1 - \mu_i)^2 \\ &= V_{i1}. \end{aligned}$$

As a conclusion we have in both cases: the distribution belongs to $\mathbb{B}_K(\boldsymbol{\mu}, \boldsymbol{\sigma})$.

**Developing the lower bound for $T_i$ in the case $V_{i1} \leq \mu_1 + \mu_i - \mu_1^2 - \mu_i^2$:** Given a joint distribution of Bernoulli variables in $\mathbb{B}_K(\boldsymbol{\mu}, \boldsymbol{V})$, let us develop the corresponding lower bound.

Let $\mathcal{A}$ be a $\delta$ sound strategy. Denote $\mathbb{P}^{(1)}$ the joint distribution of $X_i$ defined above. For $i \in \{2, \ldots, K\}$, denote by $\mathbb{P}^{(i)}$ the alternative probability distribution where only arm $i$ is modified as follows:

$$X_{i,t} = \mathbb{1}\left( \{U_t \leq a_i\} \text{ or } \left\{ W'_{i,t} \leq \frac{\mu_1 + \epsilon - \mu_i}{1 - a_i} + b_i \right\} \right),$$

where $\epsilon > 0$ and $(W'_{i,t})_{t \geq 1}$ is a sequence of independent variables following the uniform distribution in $[0, 1]$. Observe that $\mathbb{E}^{(i)}[X_i] = \mu_1 + \epsilon$, therefore under $\mathbb{P}^{(i)}$ the optimal arm is arm $i$.

Fix $i \geq 2$, since $\mathcal{A}$ is $\delta$-sound,

$$\mathbb{P}^{(1)}(\psi \neq 1) + \mathbb{P}^{(i)}(\psi \neq i) \leq 2\delta. \tag{53}$$

Using Theorem 2.2 of [29], we have

$$\mathbb{P}^{(1)}(\psi \neq 1) + \mathbb{P}^{(i)}(\psi = 1) \geq 1 - \text{TV}\left( \mathbb{P}^{(1)}, \mathbb{P}^{(i)} \right),$$

where $\text{TV}(\mathbb{P}, \mathbb{Q})$ denotes the total variation distance between $\mathbb{P}$ and $\mathbb{Q}$. Using (58), we conclude that

$$\text{TV}(\mathbb{P}^{(1)}, \mathbb{P}^{(i)}) \geq 1 - 2\delta. \tag{54}$$

For a subset $A \subset [\![K]\!]$, denote $N_A$ the total number of rounds where the jointly queried arms are the elements of $A$, and let $\mathbb{P}_A^{(\cdot)}$ denote the joint distribution of $(X_j)_{j \in A}$ under $\mathbb{P}^{(\cdot)}$. Under Protocol 1, the learner has to choose a subset $C_t \subset [\![K]\!]$ in each round $t$, and observes only the rewards of arms in $C_t$. Hence, we can apply Lemma K.5 which bounds the total variation distance in terms of Kullback-Leibler discrepancy to this case where arms correspond to subsets of $[\![K]\!]$. This gives

$$\begin{aligned} \text{TV}\left( \mathbb{P}^{(1)}, \mathbb{P}^{(i)} \right) &\leq 1 - \frac{1}{2} \exp\left\{ - \sum_{A \subset [\![K]\!]} \mathbb{E}^{(1)}[N_A] \text{KL}\left( \mathbb{P}_A^{(1)}, \mathbb{P}_A^{(i)} \right) \right\} \\ &= 1 - \frac{1}{2} \exp\left\{ - \sum_{\substack{A \subset [\![K]\!]: \\ i \in A}} \mathbb{E}^{(1)}[N_A] \text{KL}\left( \mathbb{P}_A^{(1)}, \mathbb{P}_A^{(i)} \right) \right\}. \end{aligned} \tag{55}$$

Now fix $A \subset [\![K]\!]$, such that $i \in A$. Let us calculate $\text{KL}(\mathbb{P}_A^{(1)}, \mathbb{P}_A^{(i)})$.

Denote $\mathbb{Q}_A^{(i)}$ for $i \in [\![K]\!]$, the joint distribution: $((X_{j,t})_{j \in A}, U_t)$. Using the data processing inequality

$$\text{KL}\left( \mathbb{P}_A^{(1)}, \mathbb{P}_A^{(i)} \right) \leq \text{KL}\left( \mathbb{Q}_A^{(1)}, \mathbb{Q}_A^{(i)} \right).$$

Observe that $X_1$ follows the same distribution under $\mathbb{P}^{(1)}$ and $\mathbb{P}^{(i)}$. Therefore

$$\text{KL}\left( \mathbb{P}_A^{(1)}, \mathbb{P}_A^{(i)} \right) \leq \mathbb{E}_{U_t}^{(1)}\left[ \text{KL}\left( \mathbb{P}_A^{(1)}((X_j)_{j \in A} \mid U_t), \mathbb{P}_A^{(i)}((X_j)_{j \in A} \mid U_t) \right) \right],$$

where $\mathbb{E}_{U_t}^{(\cdot)}$ denotes the conditional expectation under $\mathbb{P}^{(\cdot)}$ with respect to $U_t$.

Observe that conditionally to $U_t$, the variables $(X_j)_{j \neq 1}$ are independent, both under $\mathbb{P}_A^{(1)}$ and $\mathbb{P}_A^{(1)}$. Moreover $X_j$ for $j \in A \setminus \{1, i\}$ have the same probability distributions under both alternatives. Therefore

$$\mathrm{KL}\left(\mathbb{P}_A^{(1)}, \mathbb{P}_A^{(i)}\right) \leq \mathbb{E}_{U_t}^{(1)}\left[\mathrm{KL}\left(\mathbb{P}_A^{(1)}((X_j)_{j \in A} \mid U_t), \mathbb{P}_A^{(i)}((X_j)_{j \in A} \mid U_t)\right)\right]$$
$$= \mathbb{E}_{U_t}^{(1)}\left[\mathrm{KL}\left(\mathbb{P}_A^{(1)}(X_i \mid U_t), \mathbb{P}_A^{(i)}(X_i \mid U_t)\right)\right].$$

Therefore, we have:

$$\mathrm{KL}\left(\mathbb{P}_A^{(1)}, \mathbb{P}_A^{(i)}\right) \leq (1 - a_i)\mathrm{KL}\left(\mathbb{1}(W_{i,t} \leq b_i), \mathbb{1}\left(W_{i,t}' \leq \frac{\mu_1 + \epsilon - \mu_i}{1 - a_i} + b_i\right)\right).$$

Using Lemma K.9, we have

$$\mathrm{KL}\left(\mathbb{P}_A^{(1)}, \mathbb{P}_A^{(i)}\right) \leq (1 - a_i)\frac{\left(\frac{\mu_1 + \epsilon - \mu_i}{1 - a_i}\right)^2}{\left(\frac{\mu_1 + \epsilon - \mu_i}{1 - a_i} + b_i\right)\left(1 - \frac{\mu_1 + \epsilon - \mu_i}{1 - a_i} - b_i\right)}$$
$$= \frac{(1 - a_i)(\mu_1 + \epsilon - \mu_i)^2}{(\mu_1 + \epsilon - \mu_i + b_i - a_i b_i)(1 - (\mu_1 + \epsilon - \mu_i) - a_i - b_i + a_i b_i)}.$$

Using $a_i + b_i - a_i b_i = \mu_i$ and $a_i = \frac{\mu_i - b_i}{1 - b_i}$, we have

$$\mathrm{KL}\left(\mathbb{P}_A^{(1)}, \mathbb{P}_A^{(i)}\right) \leq \frac{(1 - \mu_i)(\mu_1 + \epsilon - \mu_i)^2}{((\mu_1 + \epsilon)(1 - b_i) - \mu_i + b_i)(1 - \mu_1 - \epsilon)}$$
$$= \frac{(1 - \mu_i)(\mu_1 + \epsilon - \mu_i)^2}{(\mu_1 + \epsilon - \mu_i + b_i(1 - \mu_1 - \epsilon))(1 - \mu_1 - \epsilon)}$$

Next, we plug the inequality above into inequality (55) and obtain:

$$\mathrm{TV}\left(\mathbb{P}^{(1)}, \mathbb{P}^{(i)}\right) \leq 1 - \frac{1}{2}\exp\left\{-\mathrm{KL}\left(\mathbb{P}_A^{(1)}, \mathbb{P}_A^{(i)}\right)\sum_{\substack{A \subset [\![K]\!]: \\ i \in A}} \mathbb{E}^{(1)}[N_A]\right\}.$$

For $j \in [\![K]\!]$, denote by $T_j$ the total number of rounds where arm $j$ was queried:

$$T_j := \sum_{\substack{A \subset [\![K]\!]: \\ j \in A}} N_A.$$

Therefore

$$\mathrm{TV}\left(\mathbb{P}^{(1)}, \mathbb{P}^{(i)}\right) \leq 1 - \frac{1}{2}\exp\left\{-\mathrm{KL}\left(\mathbb{P}_A^{(1)}, \mathbb{P}_A^{(i)}\right)\mathbb{E}^{(1)}[T_i]\right\}.$$

Combining the inequality above with (54) and taking $\epsilon \to 0$, we obtain

$$\mathbb{E}^{(1)}[T_i] \geq \frac{(\mu_1 + \epsilon - \mu_i + b_i(1 - \mu_1 - \epsilon))(1 - \mu_1 - \epsilon)}{(1 - \mu_i)(\mu_1 + \epsilon - \mu_i)^2}\log\left(\frac{1}{4\delta}\right).$$

Taking $\epsilon \to 0$, using the expression of $b_i$ and $\mu_i, \mu_1 \leq 3/4$ yields

$$\mathbb{E}^{(1)}[T_i] \geq \frac{(\mu_1 - \mu_i) + V_{i1} + (\mu_1 - \mu_i)^2}{2(\mu_1 - \mu_i)^2} \cdot \frac{1 - \mu_1}{1 - \mu_i}\log\left(\frac{1}{4\delta}\right)$$
$$\geq \frac{V_{i1} + (\mu_1 - \mu_i)}{8(\mu_1 - \mu_i)^2}\log\left(\frac{1}{4\delta}\right). \tag{56}$$

**Developing a lower bound for $T_i$ in the case $V_{i1} \geq \mu_1 + \mu_i - \mu_1^2 - \mu_i^2$.** In this case we introduce the following alternative distribution $\mathbb{P}^{(i)}$, where we only change the variable $X_i$ into: $\forall t \geq 1$ : $X_{i,t} = \mathbb{1}(W_{i,t} \leq \mu_1 + \epsilon)$, for some small positive constant $\epsilon$.

Similarly to the previous case, since all variables are independent: for any $A \subset [\![K]\!]$:

Therefore

$$\mathrm{KL}\left(\mathbb{P}_A^{(1)}, \mathbb{P}_A^{(i)}\right) = \mathrm{KL}(\mathbb{1}(W_{i,t} \leq \mu_i), \mathbb{1}(W_{i,t} \leq \mu_1 + \epsilon))$$

$$\leq \frac{(\mu_1 + \epsilon - \mu_i)^2}{(\mu_1 + \epsilon)(1 - \mu_1 - \epsilon)}$$

The remainder of the calculation is similar to the preivous case and leads to (using $\mu_i \in [1/3, 3/4]$):

$$\mathbb{E}^{(1)}[T_i] \geq \frac{\mu_1(1 - \mu_1)}{(\mu_1 - \mu_i)^2} \log\left(\frac{1}{4\delta}\right)$$

$$\geq \frac{3}{16(\mu_1 - \mu_i)^2} \log\left(\frac{1}{4\delta}\right)$$

$$\geq \frac{3(V_{i1} + \mu_1 - \mu_i)}{16(\mu_1 - \mu_i)^2} \log\left(\frac{1}{4\delta}\right), \tag{57}$$

where we used in the last line the fact that: $V_{i1} + \mu_1 - \mu_i \leq 5/4$

**Conclusion.** Using (56) and (57), we have for any $i \in \{2, \ldots, K\}$:

$$\mathbb{E}^{(1)}[T_i] \geq \frac{V_{i1} + (\mu_1 - \mu_i)}{8(\mu_1 - \mu_i)^2} \log\left(\frac{1}{4\delta}\right).$$

Recall that $T_i$ represents the number of rounds where arm $i$ is queried. Hence the total number of queries $N$ satisfies: $N \geq \sum_{i=2}^{K} T_i$. We conclude that

$$\mathbb{E}^{(1)}[N] \geq \log\left(\frac{1}{4\delta}\right) \sum_{i=2}^{K} \frac{V_{i1} + (\mu_1 - \mu_i)}{8(\mu_1 - \mu_i)^2}.$$

## J    Proof of Theorem 6.2

Without loss of generality assume that $\mu_1 > \mu_2 \geq \cdots \geq \mu_K$, hence $i^* = 1$. Let $\mathcal{A}$ be a $\delta$ sound strategy. Let $\mathbb{P}^{(1)}$ denote the joint distribution of the arms defined as follows: Let $Z$ denote a random variable distributed following $\mathcal{N}(\mu_1, 1)$.

- For $i \in \{2, \ldots, K\}$, we set $X_i = Z + Y_i$, where $Y_i$ is a random variable independent of $Z$ and $(Y_i)_{i \geq 2}$, and distributed following $\mathcal{N}(\mu_i - \mu_1, V_{i1})$.
- The optimal arm is given by $X_1 = Z$, let $\sigma_1 := 1$.

Recall that in the configuration above we have for each $i \in [\![K]\!]$: $\mathbb{E}[X_i] = \mu_i$, $\mathbb{E}[(X_1 - X_i)^2] = V_{i1}$, and $\mathrm{Var}(X_i) \geq 1$, therefore $\mathbb{P}^{(0)} \in \mathbb{G}_K(\boldsymbol{\mu}, \boldsymbol{V})$.

For $i \in \{2, \ldots, K\}$, denote by $\mathbb{P}^{(i)}$ the alternative probability distribution where the only arm modified is arm $i$. We set: $X_i = Z + Y'$, where $Y' \sim \mathcal{N}(\epsilon, V_{i1} + (\mu_1 - \mu_i)^2)$, where $\epsilon > 0$.

Observe that, under $\mathbb{P}^{(i)}$, the arm $i$ is optimal. Fix $i \geq 2$. Since $\mathcal{A}$ is $\delta$-sound, we have

$$\mathbb{P}^{(1)}(\psi \neq 1) + \mathbb{P}^{(i)}(\psi \neq i) \leq 2\delta. \tag{58}$$

Using Theorem 2.2 of [29], we have

$$\mathbb{P}^{(1)}(\psi \neq 1) + \mathbb{P}^{(i)}(\psi = 1) \geq 1 - \mathrm{TV}\left(\mathbb{P}^{(1)}, \mathbb{P}^{(i)}\right),$$

where $\mathrm{TV}(\mathbb{P}, \mathbb{Q})$ denotes the total variation distance between $\mathbb{P}$ and $\mathbb{Q}$. Using (58), we deduce that

$$\mathrm{TV}(\mathbb{P}^{(1)}, \mathbb{P}^{(i)}) \geq 1 - 2\delta. \tag{59}$$

For a subset $A \subset [\![K]\!]$, denote $N_A$ the total number of rounds where the jointly queried arms are the elements of $A$, and let $\mathbb{P}_A^{(\cdot)}$ denote the joint distribution of $(X_j)_{j \in A}$ under $\mathbb{P}^{(\cdot)}$. Under Protocol 1, the learner has to choose a subset $C_t \subset [\![K]\!]$ in each round $t$, and observes only the rewards of arms in $C_t$. Hence, we can apply Lemma K.5 which bounds the total variation distance in terms of Kullback-Leibler discrepancy to this case where arms correspond to subsets of $[\![K]\!]$. This gives

$$\begin{aligned}
\mathrm{TV}\left(\mathbb{P}^{(1)}, \mathbb{P}^{(i)}\right) &\leq 1 - \frac{1}{2} \exp\left\{ -\sum_{A \subset [\![K]\!]} \mathbb{E}^{(1)}[N_A] \mathrm{KL}\left(\mathbb{P}_A^{(1)}, \mathbb{P}_A^{(i)}\right) \right\} \\
&= 1 - \frac{1}{2} \exp\left\{ -\sum_{\substack{A \subset [\![K]\!]: \\ i \in A}} \mathbb{E}^{(1)}[N_A] \mathrm{KL}\left(\mathbb{P}_A^{(1)}, \mathbb{P}_A^{(i)}\right) \right\}. \tag{60}
\end{aligned}$$

Now fix $A \subset [\![K]\!]$, such that $i \in A$. Let us calculate $\mathrm{KL}(\mathbb{P}_A^{(1)}, \mathbb{P}_A^{(i)})$.

Using the data processing inequality, we deduce that, for any $A \subset [\![K]\!]$,

$$\mathrm{KL}\left(\mathbb{P}_A^{(1)}, \mathbb{P}_A^{(i)}\right) \leq \mathrm{KL}\left(\mathbb{P}_{A \cup \{1\}}^{(1)}, \mathbb{P}_{A \cup \{1\}}^{(i)}\right).$$

Observe that $X_1$ follows the same distribution under $\mathbb{P}^{(1)}$ and $\mathbb{P}^{(i)}$. Therefore

$$\mathrm{KL}\left(\mathbb{P}_A^{(1)}, \mathbb{P}_A^{(i)}\right) \leq \mathbb{E}_{X_1}^{(1)}\left[\mathrm{KL}\left(\mathbb{P}_A^{(1)}((X_j)_{j \in A} \mid X_1), \mathbb{P}_A^{(i)}((X_j)_{j \in A} \mid X_1)\right)\right],$$

where $\mathbb{E}_{X_1}[.]$ refers to the conditional expectation with respect to $X_1$. Recall that conditionally to $X_1$, the variables $(X_j)_{j \neq 1}$ are independent, both under $\mathbb{P}_A^{(1)}$ and $\mathbb{P}_A^{(1)}$. Moreover $X_j$ for $j \in A \setminus \{1, i\}$ have the same probability distributions under both alternatives. Therefore, we can the above Kullback divergence in terms of conditional Kullback divergence.

$$\begin{aligned}
\mathrm{KL}\left(\mathbb{P}_A^{(1)}, \mathbb{P}_A^{(i)}\right) &\leq \mathbb{E}_{X_1}^{(1)}\left[\mathrm{KL}\left(\mathbb{P}_A^{(1)}((X_j)_{j \in A} \mid X_1), \mathbb{P}_A^{(i)}((X_j)_{j \in A} \mid X_1)\right)\right] \\
&= \mathbb{E}_{X_1}^{(1)}\left[\mathrm{KL}\left(\mathbb{P}_A^{(1)}(X_i \mid X_1), \mathbb{P}_A^{(i)}(X_i \mid X_1)\right)\right].
\end{aligned}$$

Moreover, we have

$$\begin{aligned}
\mathbb{E}_A^{(1)}[X_i \mid X_1] &= \mu_i + X_1 - \mu_1 \\
\mathbb{E}_A^{(i)}[X_i \mid X_1] &= \mu_1 + \epsilon + X_1 - \mu_1 \\
\mathrm{Var}_A^{(1)}(X_i \mid X_1) &= 1 + V_{i1} \\
\mathrm{Var}_A^{(i)}(X_i \mid X_1) &= 1 + V_{i1} + (\mu_1 - \mu_i)^2.
\end{aligned}$$

We deduce that

$$\begin{aligned}
\mathrm{KL}\left(\mathbb{P}_A^{(1)}, \mathbb{P}_A^{(i)}\right) &\leq \frac{1}{2} \log\left(\frac{1 + V_{i1} + (\mu_1 - \mu_i)^2}{1 + V_{i1}}\right) + \frac{1 + V_{i1} + (\mu_i - \mu_1 - \epsilon)^2}{2(1 + V_{i1})} - \frac{1}{2} \\
&\leq \frac{1}{2} \log\left(\frac{V_{i1} + (\mu_1 - \mu_i)^2}{V_{i1}}\right) + \frac{(\mu_i - \mu_1 - \epsilon)^2 - (\mu_i - \mu_1)^2}{2(1 + V_{i1} + (\mu_1 - \mu_i)^2)}.
\end{aligned}$$

As a conclusion, we have

$$\mathrm{KL}\left(\mathbb{P}_A^{(1)}, \mathbb{P}_A^{(i)}\right) \leq \frac{1}{2} \log\left(\frac{V_{i1} + (\mu_1 - \mu_i)^2}{V_{i1}}\right) + \frac{(\mu_i - \mu_1 - \epsilon)^2 - (\mu_i - \mu_1)^2}{2(1 + V_{i1} + (\mu_1 - \mu_i)^2)}. \tag{61}$$

Next, we plug the inequality above into (60) and obtain:

$$\mathrm{TV}\Big(\mathbb{P}^{(1)},\mathbb{P}^{(i)}\Big) \leq 1 - \frac{1}{2}\exp\left\{-\mathrm{KL}\Big(\mathbb{P}^{(1)}_A,\mathbb{P}^{(i)}_A\Big)\sum_{\substack{A\subset[\![K]\!]: \\ i\in A}}\mathbb{E}^{(1)}[N_A]\right\}.$$

For $j\in[\![K]\!]$, denote by $T_j$ the total number of rounds where arm $j$ was queried:

$$T_j := \sum_{\substack{A\subset[\![K]\!]: \\ j\in A}}N_A.$$

Therefore

$$\mathrm{TV}\Big(\mathbb{P}^{(1)},\mathbb{P}^{(i)}\Big) \leq 1 - \frac{1}{2}\exp\Big\{-\mathrm{KL}\Big(\mathbb{P}^{(1)}_A,\mathbb{P}^{(i)}_A\Big)\mathbb{E}^{(1)}[T_i]\Big\}.$$

Combining the inequality above with (59) and taking $\epsilon\to 0$, we obtain

$$\mathbb{E}^{(1)}[T_i] \geq \frac{2}{\log\Big(\frac{V_{i1}+(\mu_1-\mu_i)^2}{V_{i1}}\Big)}\log\Big(\frac{1}{4\delta}\Big).$$

Taking $\epsilon\to 0$ and using the definition of $\sigma_i$ for $i\geq 2$, we have

$$\mathbb{E}^{(1)}\left[\sum_{i=2}^{K}T_i\right] \geq 2\log\Big(\frac{1}{4\delta}\Big)\sum_{i=2}^{K}\frac{1}{\log\Big(\frac{V_{i1}+(\mu_1-\mu_i)^2}{V_{i1}}\Big)}.$$

Recall that $T_i$ represents the number of rounds where arm $i$ is queried. Hence the total number of queries $N$ satisfies: $N\geq\sum_{i=2}^{K}T_i$. We conclude that

$$\mathbb{E}^{(1)}[N] \geq 2\log\Big(\frac{1}{4\delta}\Big)\sum_{i=2}^{K}\frac{1}{\log\Big(1+\frac{(\mu_1-\mu_i)^2}{V_{i1}}\Big)}.$$

## K  Some technical results

We state below a version of the empirical Bernstein's inequality presented in [3].

**Theorem K.1.** *Let $X_1,\ldots,X_t$ be i.i.d random variables taking their values in $[0,b]$. Let $\mu=\mathbb{E}[X_1]$ be their common expected value. Consider the empirical expectation $\bar{X}_t$ and variance $V_t$ defined respectively by*

$$\bar{X}_t = \frac{\sum_{i=1}^{t}X_i}{t} \quad \text{and} \quad V_t = \frac{\sum_{i=1}^{t}(X_i-\bar{X}_t)^2}{t}.$$

*Then for any $t\in\mathbb{N}$ and $x>0$, with probability at least $1-3e^{-x}$*

$$\big|\bar{X}_t - \mu\big| \leq \sqrt{\frac{2V_t x}{t}} + \frac{3bx}{t}.$$

Theorem below corresponds to Theorem 10 in [24].

**Theorem K.2.** *Let $n\geq 2$ and $\boldsymbol{X}=(X_1,\ldots,X_n)$ be a vector of independent random variables with values in $[0,1]$. Then for $\delta>0$ we have, writing $\mathbb{E}V_n$ for $\mathbb{E}_{\boldsymbol{X}}V_n(\boldsymbol{X})$,*

$$\mathbb{P}\left\{\sqrt{\mathbb{E}V_n} > \sqrt{V_n(\boldsymbol{X})} + \sqrt{\frac{2\log(1/\delta)}{n-1}}\right\} \leq \delta$$

$$\mathbb{P}\left\{\sqrt{V_n(\boldsymbol{X})} > \sqrt{\mathbb{E}V_n} + \sqrt{\frac{2\log(1/\delta)}{n-1}}\right\} \leq \delta.$$

The following lemma is technical, it will be used in the proof of Lemma E.6.

**Lemma K.3.** *Let $a, b, c$ and $d > 0$, we have*

$$\frac{a+b}{c+d} \leq \max\left\{\frac{a}{c}, \frac{b}{d}\right\}.$$

*Proof.* Let $\rho = \frac{c}{c+d} \in (0,1)$. Observe that

$$\frac{a+b}{c+d} = \rho\frac{a}{c} + (1-\rho)\frac{b}{d},$$

and $1 - \rho = \frac{d}{c+d} \in (0,1)$. Taking the maximum of the convex combination above gives the result. $\qquad\square$

**Lemma K.4.** *Let $x \geq 1, c \in (0,1)$ and $y > 0$ such that:*

$$\frac{\log(x/c)}{x} > y. \tag{62}$$

*Then:*

$$x < \frac{2\log\left(\frac{1}{cy}\right)}{y}.$$

*Proof.* Inequality (62) implies

$$x < \frac{\log(x/c)}{y},$$

and further

$$\log(x/c) < \log(1/yc) + \log\log(x/c) \leq \log(1/yc) + \frac{1}{2}\log(x/c),$$

since it can be easily checked that $\log(t) \leq t/2$ for all $t > 0$. Solving and plugging back into the previous display leads to the claim. $\qquad\square$

**Lemma K.5** (20, with slight modification). *Let $\nu$ and $\nu'$ be two collections of $d$ probability distributions on $\mathbb{R}$, such that for all $a \in [\![d]\!]$, the distributions $\nu_a$ and $\nu_{a'}$ are mutually absolutely continuous. For any almost-surely finite stopping time $\tau$ with respect to $(\mathcal{F}_t)$,*

$$\sup_{\mathcal{E} \in \mathcal{F}_\tau} |\mathbb{P}_\nu(\mathcal{E}) - \mathbb{P}_{\nu'}(\mathcal{E})| \leq 1 - \frac{1}{2}\exp\left\{-\sum_{a=1}^{d}\mathbb{E}_\nu[N_a(\tau)]KL(\nu_a, \nu'_a)\right\}.$$

**Lemma K.6.** *[30] Let $(Y_1, \ldots, Y_n)$ be i.i.d Gaussian variables, with mean $0$ and variance $1$. Let $Z = \sum_{i=1}^{n} Y_i^2$. For any number $0 < x < 1$,*

$$\mathbb{P}\left(Z \geq n - 1 + 2\sqrt{(n-1)\log(1/x)} + 2\log(1/x)\right) \leq x,$$

$$\mathbb{P}\left(Z \leq n - 1 - 2\sqrt{(n-1)\log(1/\delta)}\right) \leq x.$$

*For any positive number $0 < x < 1$*

$$\mathbb{P}\left(Z \leq (n-1)Cx^{2/(n-1)}\right) \leq x,$$

*where the constant $C = \exp(-1)$.*

**Concentration bound for the Gaussian variance sample.** Let $\boldsymbol{X} = (X_1, \ldots, X_n)$ be a vector of independent standard normal variables. Define the sample variance by

$$V_n(\boldsymbol{X}) := \frac{1}{n(n-1)}\sum_{1 \leq i < j \leq n}(X_i - X_j)^2. \tag{63}$$

Observe that

$$V_n(\boldsymbol{X}) = \frac{1}{2n(n-1)} \sum_{i \neq j} (X_i - X_j)^2$$

$$= \frac{1}{n(n-1)} \left( (n-1) \sum_{i=1}^{n} X_i^2 - \sum_{i \neq j} X_i X_j \right)$$

$$= \frac{1}{n(n-1)} \boldsymbol{X}^\top A \boldsymbol{X},$$

where $A$ is the matrix such that off-diagonal entries are equal to $-1$ and diagonal entries are equal to $n-1$. Let us compute the eigenvalue of the matrix $A$: Observe that $A = n\boldsymbol{I}_n - \mathbb{1}_n\mathbb{1}_n^\top$, hence the eigenvalues of $A$ are $n$ with multiplicity $n-1$ and $0$. Hence, we have

$$\boldsymbol{X}^\top A \boldsymbol{X} = n \sum_{i=1}^{n-1} Y_i^2,$$

where $(Y_i)$ are independent and follow the standard normal distribution. Finally using Lemma K.6 we obtain:

**Lemma K.7.** *Let $\boldsymbol{X} = (X_1, \ldots, X_n)$ be a vector of independent standard normal variables. Let $V_n(\boldsymbol{X})$ denote the variance sample defined in* (63)*. Let $\delta \in (0, 1/3)$, we have with probability at least $1 - 3\delta$:*

$$V_n(\boldsymbol{X}) \geq \max\left\{ 1 - 2\sqrt{\frac{\log(1/\delta)}{n-1}}; C\delta^{2/(n-1)} \right\}$$

$$V_n(\boldsymbol{X}) \leq 1 + 2\sqrt{\frac{\log(1/\delta)}{n-1}} + 2\frac{\log(1/\delta)}{n-1}.$$

**Lemma K.8.** *Let $X$ and $Y$ be two Bernoulli variables with means $x$ and $y$ respectively. We have*

$$\max(x + y - 1, 0) \leq \mathbb{E}[XY] \leq \min(x, y).$$

*Moreover:*

$$x - y - (x-y)^2 \leq \mathrm{Var}(X - Y) \leq \min(2 - (x+y); \ x+y) - (x-y)^2.$$

*Proof.* Without loss of generality suppose that $x \leq y$. We have

$$\mathbb{E}[XY] = \mathbb{P}(XY = 1)$$
$$= \mathbb{P}(X = 1 \quad \text{and} \quad Y = 1)$$
$$= \mathbb{P}(X = 1) + \mathbb{P}(Y = 1) - \mathbb{P}(X = 1 \quad \text{or} \quad Y = 1)$$
$$= x + y - \mathbb{P}(X = 1 \quad \text{or} \quad Y = 1).$$

The conclusion follows by using $y \leq \mathbb{P}(X = 1 \quad \text{or} \quad Y = 1) \leq 1$, and $XY \geq 0$.

Moreover, we have:

$$\mathrm{Var}(X - Y) = \mathbb{E}\big[(X - Y)^2\big] - (x - y)^2$$
$$= \mathbb{E}[X] + \mathbb{E}[Y] - 2\mathbb{E}[XY] - (x - y)^2$$
$$= x + y - (x - y)^2 - 2\mathbb{E}[XY].$$

We plug-in the previous bounds on $\mathbb{E}[XY]$ and obtain the result. $\qquad\square$

**Lemma K.9.** *Let $X$ and $Y$ denote two Bernoulli variables with paramters $x \in (0, 1)$ and $y \in (0, 1)$ respectively. We have*

$$KL(X, Y) \leq \frac{(x - y)^2}{y(1 - y)}.$$

*Proof.* Using $\log u \leq u - 1$ we get

$$\mathrm{KL}(X, Y) = x \log \frac{x}{y} + (1 - x) \log \frac{1 - x}{1 - y} \leq \frac{x^2}{y} + \frac{(1 - x)^2}{1 - y} - 1 = \frac{(x - y)^2}{y(1 - y)}.$$

$$\square$$