# OpenReview forum: "Covariance-adaptive best arm identification"
_NeurIPS.cc/2023/Conference — NeurIPS 2023 poster_

### Official Review · Reviewer_XxU8 · 2023-06-21

**Soundness:** 3 good
**Presentation:** 3 good
**Contribution:** 3 good
**Rating:** 7
**Confidence:** 3

**Summary:**

This work focuses on best arm identification, that is, determining the arm with highest average return within a small number of samples (sample complexity). The twist is that authors propose a framework where sets of arms (of arbitrary size) can be queried at the same time, and the set of corresponding rewards is observed. This setting goes beyond classical frameworks, as observations in a set of simultaneously queried arms are dependent. Moreover, the arm pairwise covariances are unknown to the bandit algorithm. Authors propose an algorithm tackling this problem, which nearly matches associated lower bounds, and empirically evaluate it.

**Strengths:**

- Originality: Authors introduce a novel best arm identification setting which might have interesting real-life applications. Differences with prior works are thoroughly discussed and the discussion covers a large set of related settings.
- Quality: The results in this paper seem technically sound, although I did not check the proofs in detail. The experimental study is convincing and supports theoretical claims. The code is available, which helps reproducibility and potential future follow-up.
- Significance: The setting tackled in this paper introduces new technical challenges (in particular to derive lower bounds), and the technical tools used in proofs can be of independent interest.

**Weaknesses:**

- Clarity: I appreciated the many examples provided in order to illustrate intuitions behind the papers, although I believe the section from Line 59 to Line 123 is a bit messy and hard to follow. Adding more structure to this section and clearly highlighting the different points covered in that paragraph would really help.
- Significance: The number of rounds after which an arm can be safely eliminated from the set of “queriable” arms seems really large and not very practical. However, this weakness has been identified in this paper and properly discussed (notably by providing a version of the upper bound when elimination happens when an arm is not more considered a candidate).

**Questions:**

- What would be the main hurdles in extending the lower bound theorems to any one-dimensional exponential family?

**Limitations:**

Authors have adequately addressed the issue related to the number of rounds after which an arm is completely eliminated from future queries. This paper deals with very theoretical guarantees, and, as such, do not present significant direct negative societal impacts.

---

> ### Author Rebuttal · Authors · 2023-08-09
>
> We thank the reviewer for the valuable feedback.
>
> In the context of developing lower bounds, the conventional approach involves creating two distributions of the arms that are identical except for one arm, which is made optimal in one of the distributions. Subsequently, a lower bound is established on the number of queries necessary to differentiate between these distributions, yielding a lower bound for the number of queries for the specific arm i. However, this method proves insufficient for deriving precise lower bounds in our particular scenario. As an alternative direction, we propose exploring the creation of alternative distributions in which all arms are modified instead of just one.

---

> > ### Comment · Reviewer_XxU8 · 2023-08-14
> > **Response to the rebuttal**
> >
> > I thank the authors for their global rebuttal and their answer to my question. As my score is already positive, with respect to potential impact, I will keep the score as it is for now.

---

### Official Review · Reviewer_pLAq · 2023-07-04

**Soundness:** 3 good
**Presentation:** 2 fair
**Contribution:** 3 good
**Rating:** 6
**Confidence:** 4

**Summary:**

Authors investigate the stochastic Best Arm Identification (BAI) problem when the arms have an unknown correlation structure.
In contrast, the majority of BAI literature focuses on independent arms with unknown variances.
Authors propose algorithms for BAI in two settings (bounded random variables and gaussian random variables) and analyze the sample complexity of these algorithms.
In order to leverage the correlation structure, multiple arms are allowed to be sampled at each round.
They show that improved sample complexity can be achieved by estimating the correlation structure; moreover, the sample complexity is within a constant factor of the sample complexity for independent arms (even when independence is unknown in advance).
Simulations are run to demonstrate the method on a synthetic dataset.

**Strengths:**

The main strength of the paper is an investigation into the stochastic BAI problem where independence between the arms is not assumed.
Authors develop algorithms where the covariance structure is estimated and used to help quickly eliminate suboptimal arms.
Of course, multiple arms must be sampled simultaneously in order to estimate the covariance between them.
On one hand, the analysis (and simulations) show that nice speed-ups in sample complexity can be obtained when the covariance structure is taken into account.
On the other hand, the analysis seems to indicate that the asymptotic rate of the sample complexity does not increase even when the arms are independent (but this dependence is unknown).
This has exciting implications for a variety of real-world settings where arms may be expected to be correlated.

**Weaknesses:**

There are two main weaknesses of the paper.

The first main weakness is that the paper seems to not discuss (and sometimes ignore?) what I would consider to be a very important consideration: the "overhead cost" of estimating the covariance.
It seems intuitive that a method which places no assumptions on the covariances -- and therefore must estimate them -- must incur some overhead cost when compared to a setting where there are structured assumptions on the covariance (e.g. the independent setting).
This overhead cost may not be very large, but I am pretty certain that it should exist.
A couple of times throughout the paper (Lines 103, 247) authors claim that the sample complexity bounds they obtain are smaller than the independent case; however, it seems that they are smaller than a constant factor (say, 2) of the sample complexity of the independent case.
In terms of the asymptotic rate of the sample complexity, this doesn't matter; however, in terms of finite sample, practical performance, this does seem to matter and it points out exactly this "overhead cost".
Overall, I think that the paper would be stronger if authors more appropriately discussed their results in this context, i.e. the existence of the overhead cost of estimating covariance relative to assuming it, but with large gains when dependencies help.

The second weakness of the paper is that the exact nature of the guarantees on the sample complexity are not so clear.
From what I understand, these algorithms need to contain the best arm with probability $1 - \delta$ for some total number of arm queries $N$ and this stopping time is a random variable.
The goal of the algorithm designer is that, under appropriate conditions, $N$ is small in some sense, as a random variable.
So what does it mean then in Theorems 4.1 and 5.1 that authors write $N \leq XYZ$? Does this happen with probability 1 (that seems unlikely)?
Does this mean that $N$ is order $XYZ$ in probability?
Or is $N$ not a random variable?
Perhaps there is some convention in the BAI literature so that this is very clear; but to someone in adjcent fields, this could use clarification.
I'm hopeful that this can be clarified and the fix is relatively easy.

Further comments on clarity and discussion of results are given below.

### Minor Comments

1. (Line 58): What is the $\pi$ subscript in $N$ represent and why does it never appear again in the paper? Moreover, the last sentence in this paragraph seems to be a sentence fragment that is missing a part.
2. (Line 64): Could you describe more precisely what "comparing the means" is referring to here? What is the precise statistical problem i.e. estimation, hypothesis testing etc ?
3. (Line 71): A motivating example of clinical trials is discussed, "whre the effects of drugs on patients with similar traits or comparing drugs with similar components may exhibit underlying correlations". However, the correlation between counterfactual outcomes (arms) can't be simultaneosuly observed in a causal inference setting -- only one outcome may be observed for each patient, often known as the "fundamental problem of causal inference". I recommend rephrasing this motivation so that arms cannot be interpreted to be counterfactual (and thus not simulatenously observable) outcomes, or selecting a different motivating example.
4. (line 103): authors write that "quantity (3) is always smaller (up to a numerical constant) than its independent case counterpart". I think this is phrased in an unfortunately misleading way. I presume that the authors mean "quantity (3) is no larger than a constant times the independent case counterpart". Unfortunately, writing the phrase "always smaller" gives the wrong impression that quantity (3) is at most quantity (2), which does not seem to be true. Great care should be given to statements like this, so they are not misrepresenting what authors have shown.
5. (Line 140, 157) citations appear in parathensis when other citations appear in brackets. A consistent stlye should be used throughout the paper.
6. (Line 142) "The objective is to maximize the cumulative regret..." is this a typo, i.e. "maximize" should be "minimize"?
7. (Line 172): The estimated quantities $V_{i,j,t}$ and $\mu_{i,t}$ should be defined (markdown doesn't like the hat notation, sorry for dropping it). In particular, these quantities are perhaps ill-defined without reference to the algorithm: how should we interpret the sample covariance up to iteration $t$ if arm $i$ was sampled at some round $s \leq t$ but $j$ was not?
8. (Line 7 of Algorithm 2): "Jointly query all the experts in $C$". The terminology "experts" is not defined, so I'd recommend not using it here.
9. (Line 247): Authors write "the sample complexity is on the order $X$ which is larger than both our bounds (5) and (6)". If I understand correctly, it seems that the correct statement is that "(5) and (6) are less than 2 $X$". It seems very important for authors to correctly characterize their contributions so readers do not misinterpret them.
10. (Line 275): the reference (20) is perhaps misplaced.
11. (Line 305): "$\delta$-sound algorithm" is used but not defined. From context clues, I think I understand, but authors should clarify it for the readers.
12. (Line 309): Authors claim that "Algorithm 2 is nearly optimal" but this is up to a $\log(K)$ and $\log(\Lambda)$ factors. I understand $K$, but I think that $\log(\Lambda)$ deserves some discussion. How large should we expect this to be?
13. (Line 319): "Theorem 6.2 demonstrates that our algorithm achieves near-optimal performance...up to a logarithmic factor". Same as above.
14. (Line 329): Authors write that "We stress that both variants guarantee a $\delta$-sound decision on the optimal arm" but it seems that reducing $82 t$ to $2 t$ actually doesn't necessarily satisfy $\delta$-sound? If I am correct in this understanding, then the claim should be modified appropriately.
15. (Line 356): Authors write that there is "an algorithm that compares candidate arms with convex combinations of the remaining arms" but this is not discussed in the main body nor in the simulations. This seems it requires more discussion to better contextualize it within the current work or be removed all together.
16. (Line 347): Authors write "In both experiments, we observe that Pairwise-BAI+ performs worse compared to Pairwise-BAI, indicating that, empirically, in the given scenarios, continuing to sample sub-optimal arms does not contribute to improved performance". There is either a typo here or I am a bit confused. I thought that Pairwise-BAI+ was where the number of extra rounds was reduced, so if it performs worse wouldn't that indicate that we need those extra rounds? I think there is a typo, but I want to bring this to the attention of the authors to confirm or deny.

**Questions:**

I think this paper makes some interesting and strong contributions.
Unfortunately, I found some of the results and their discussions to be sufficiently confusing to a broader audience that I have set my score to "borderline reject".
However, if authors can answer the following questions to clarify these points (especially the first two), then I am more than happy to raise my score.

1. Can you please respond to my comments on the confusion regarding what $N \leq XYZ$ means if $N$ is a random variable? I'm hopeful that you can clarify this for me.
2. Can you please respond to my comments on the overhead cost of estimating the covariance structure and whether my assessment that the sample complexity bounds are within a factor of two of the independent case is correct?
3. Can Assumption 3 be relaxed to subgaussian? If the answer is no, then why not?
4. Can you comment more on the "algorithm that compares candidate arms with convex combinations of the remaining arms"? Why does it not appear in simulations and does it offer any additional benefits over the proposed algorithms?

**Limitations:**

yes

---

> ### Author Rebuttal · Authors · 2023-08-09
>
> We thank the reviewer for the valuable feedback.
> * **On Question 1:** about the formulation of our results. We apologise for not being clear. Please note that such presentation of the guarantees is standard in the literature of best arm identification with fixed confidence, see for instance: Karnin et al. (2013) and Jamieson et al. (2014). Allow us to clarify that the total number of queries, $N$, is a random variable depending on the learner's strategy and the observed samples. The guarantees we present in Theorems 4.1 and 5.1 take the form of an upper bound on the variable $N$, which holds with a probability of at least $1-\delta$. The statement of these theorems says that with probability at least $1-\delta$: [the algorithm finds the best arm --and-- the total number of queries satisfies $N\le f(\delta)$].
> * **On Question 2:** Regarding the overhead cost of unknown covariances, we acknowledge that there two different costs depending on the variables' distributions:
>    + For bounded variables: If variances were known, we could carry the same analysis using the standard Bernstein inequality (exposed in Boucheron et al 2013, Theorem 2.10) with exact variances. However, since variances are not known, we use the empirical Bernstein inequality (Maurer and Pontil 2009), in which the true variance is replaced by its empirical counterpart. The (accuracy) cost when we plug the empirical estimate appears in the numerical constant of the sub-exponential tail term (the $\mathcal{O}(1/n)$ term) where we obtain a constant of $7/3$ instead of $1/3$ in the standard Bernstein version. This cost, as pointed out by the reviewer, is merely a numerical factor when compared to the sample complexity of algorithms designed explicitly for independent arms. On the other hand, when arms are dependent, our guarantee demonstrates that being adaptive to covariance can result in a substantial improvement, possibly by an arbitrary factor, in certain scenarios as highlighted in the introductory examples in lines 78-107.
>   + For Gaussian variables the overhead of unknown variances takes a different form: if the sample size is larger than  $\log(1/\delta)$, the cost of plugging in the empirical variance estimate into the Chernoff's concentration inequality is only a multiplicative constant slightly larger than one (nearly $1+2\sqrt{\log(1/\delta)/n}$). However, in the case of a small sample regime ($n<\log(1/\delta)$), the cost is a multiplicative factor of $\exp(\sqrt{\log(1/\delta)/n}+1/2)$ due to the nature of the left tail of the chi-squared distribution (we refer the reviewer to the Sections D and K of the appendix for detailed calculations). We would like to draw the reviewer's attention that, for most natural regimes, the number of queries made for each arm is larger than $\log(1/\delta)$, hence the last described effect does not arise. However, in some specific regimes (such as the case of very small variances of the arms) an optimal algorithm should query less than $\log(1/\delta)$ samples, which necessitates introducing the exponential multiplicative term presented above into the concentration upper bound. This translates into a different form of guarantee presented in Theorem 5.1, inequality (8). It is important to note the cost in this regime cannot be avoided as highlighted by our lower bound presented in Theorem 6.2.
>   + In summary, the overhead cost of unknown covariance is rather mild except in very specific regimes.
> * **On Question 3:** Regarding the extension to sub-Gaussian variables, we would like to clarify that our algorithm relies on the empirical Bernstein inequality, which was originally designed in the literature for bounded variables. However, we have extended this inequality to accommodate Gaussian variables by leveraging existing concentration results. It is essential to note that developing such inequalities for sub-Gaussian variables is generally a non-trivial task. One possible direction to extend the considered class of distributions is to suppose that arms follow a sub-Gaussian distribution and satisfy a Bernstein moment assumption (such extensions were pointed out by works on bounded variables e.g., Balsubramani and Ramdas 2016). Given the last class of function, we can build on the standard Bernstein inequality with known variance, then plug-in an estimate of the empirical variance leveraging the concentration of quadratic forms (see Bellec 2019). However, it remains uncertain whether an extension for sub-Gaussian variables (without additional assumptions) is practically feasible.
> * **On Question 4:** The algorithm that compares candidate arms with convex combinations of the remaining arms is developed in Section B of the appendix. It was not included in the main body due to the limitation on the maximum number of pages allowed. In this algorithm, we compare candidate variables with a convex combination of other variables. In some cases, it can lead to improved guarantees. We refer the reviewer to the paragraph between lines 464 and 471 in the appendix for the motivation behind introducing this algorithm.
> * We thank the reviewer for pointing out the minor remarks and identifying typos. We will certainly consider and address these observations in the final version.
>
> Z. Karnin, T. Koren, and O. Somekh. Almost optimal exploration in multi-armed bandits. ICML, 2013.
>
> K. Jamieson, M. Malloy, R. Nowak, and S. Bubeck. lil’UCB: an optimal exploration algorithm for multi-armed bandits. COLT, 2014.
>
> Balsubramani and Ramdas, Sequential nonparametric testing with the law of the iterated logarithm, UAI 2016.
>
> P. C. Bellec. Concentration of quadratic forms under a Bernstein moment assumption. arXiv preprint 2019.
>
> Boucheron S., Lugosi G.  Massart P. (2013). Concentration Inequalities: A Nonasymptotic
> Theory of Independence. Univ. Press, Oxford.
>
> Andreas Maurer and Massimiliano Pontil. Empirical Bernstein bounds and sample variance penalization, 2009.

---

> > ### Comment · Reviewer_pLAq · 2023-08-11
> > **response to authors**
> >
> > I thank the authors for their thoughtful response to my review.
> >
> > **Question 1**:
> > I thank author for their answer.
> > This confusion seems to be entirely mine.
> >
> > **Question 2**:
> > I completely acknowledge that the overhead cost can be mild in typical cases, especially with strong correlations. It just seems like it must be there in the "worst case". By reading other responses, I think authors agree. It would be good to be more transparent in the paper about this.
> >
> > **Question 3**: thanks for the helpful elaboration!
> >
> > **Question 4**: thanks.
> >
> > I will raise my score to reflect the updated answers.

---

### Official Review · Reviewer_19g5 · 2023-07-05

**Soundness:** 3 good
**Presentation:** 3 good
**Contribution:** 3 good
**Rating:** 6
**Confidence:** 3

**Summary:**

This paper considers the problem of best arm identification with covariance in the fixed confidence setting, where arms can be dependent and rewards can be sampled simultaneously. The authors design algorithms that adapt to the unknown covariance of arms and prove that substantial improvement can be achieved over the standard setting. The authors also provide lower bounds and experimental results that support their theoretical findings.

**Strengths:**

1.	The considered problem is interesting and well-motivated.
2.	The theoretical analysis looks sound. Both upper and lower bounds are provided.


**Weaknesses:**

1.	What is the additional novelty and contribution of Theorem 5.1 compared to Theorem 4.1. It seems that there is no significant difference between the algorithm design and analysis for the bounded reward setting and that for the Gaussian reward setting. Why not unify them, e.g.,  present this work in the unified sub-Gaussian setting? Please correct me if my understanding is wrong.
2.	Could you compare Theorems 4.1 and 5.1with the results of existing covariance-adaptive bandit works?


**Questions:**

Please see the weaknesses above.

**Limitations:**

Please see the weaknesses above.

---

> ### Author Rebuttal · Authors · 2023-08-09
>
> We thank the reviewer for the valuable feedback.
> * **On Question 1 (part 1):** Comparison of Theorems 4.1 and 5.1: Theorem 4.1 deals with bounded variables. Here, the sum of bounded variables may exhibit a sub-exponential tail, which leads to the additional $1/(\mu_i-\mu_j)$ term in the complexity of comparing two variables. This additional term does not appear for Gaussian variables in Theorem 5.1. Hence the guarantees for the two classes of variables are different.
> * **On Question 1 (part 2):** Recall that we provided an additional guarantee (that was proved specifically for Gaussian variables) in Theorem 5.1 inequality (8), which is sharper than inequality (7) in the regime where variances are very small.
> * **On Question 1 (part 3):** Regarding the extension to sub-Gaussian variables, we would like to clarify that our algorithm relies on the empirical Bernstein inequality, which was originally designed in the literature for bounded variables. However, we have extended this inequality to accommodate Gaussian variables by leveraging existing concentration results. As noted previously, the result in this setting has a somewhat neater form, since it eschews the additional sub-exponential term. It is essential to note that developing such inequalities for sub-Gaussian variables is generally a non-trivial task. One possible direction to extend the considered class of distributions is to suppose that arms follow a sub-Gaussian distribution and satisfy a Bernstein moment assumption (such extensions were pointed out by works on bounded variables e.g., Balsubramani and Ramdas 2016). Given the last class of function, we can build on the standard Bernstein inequality with known variance, then plug in an estimate of the empirical variance leveraging the concentration of quadratic forms --see Bellec (2019). However, it remains uncertain whether an extension for sub-Gaussian variables (without additional assumptions) is practically feasible.
> * **On Question 2 (part 1):** about comparison with existing covariance-adaptive bandits works. While covariance-adaptive approaches have been explored in the context of combinatorial semi-bandits, it is important to note that the setting and objectives of this class of problems differ from ours. In our setting (Protocol 1), the learner has the flexibility to query any subset of arms, and the ultimate goal is to identify the single best arm. On the other hand, in combinatorial semi-bandits, the learner selects a subset from a predetermined set of subsets $\mathcal{M} = \left\lbrace M_1, \dots, M_d \right\rbrace$ at each round $t$, where $M_i \subset \[K\]$ and $\lVert M_i \rVert_1=m$ for a specific problem parameter $m$. However, the objective in the latter case is not to identify the best arm individually but rather to find the subset $M_i$ with the largest sum of rewards. Given these differences, we refrained from comparing the guarantees obtained in our work with those of combinatorial semi-bandits. The dissimilarities in problem structures and primary objectives lead us to believe that such a comparison would not be meaningful.
> * **On Question 2 (part 2):** As mentioned in the related work section, various works in the literature have focused on developing strategies that adapt to the variances of the arms. In contrast, our guarantees go beyond being solely adaptive to arm variances; they also incorporate adaptivity to the covariances between different arms
> In lines 242-248, we have provided a comparison of these two types of guarantees. Specifically, when arms are independent, our guarantees exhibit adaptivity to variances, though with a numerical factor of $2$. However, when arms are correlated, the improvement achievable with our procedure can be an arbitrary factor, favoring our approach.
>
> Balsubramani and Ramdas, Sequential nonparametric testing with the law of the iterated logarithm, UAI 2016.
>
> P. C. Bellec. Concentration of quadratic forms under a Bernstein moment assumption. arXiv preprint 2019.

---

> > ### Comment · Reviewer_19g5 · 2023-08-15
> > **Thank the authors for their response**
> >
> > Thank the authors for their response. My concerns were addressed. I raised my score from 5 to 6.

---

### Official Review · Reviewer_ZvwR · 2023-07-06

**Soundness:** 3 good
**Presentation:** 3 good
**Contribution:** 2 fair
**Rating:** 5
**Confidence:** 4

**Summary:**

This paper focuses on the question of identifying $\epsilon$-optimal arms given a confidence input $\delta$, or in other words, under the PAC model. Instead of pulling only one arm and observing the rewards, the authors leverage the underlying structure of arm distributions by allowing multiple queries per round (Protocol 1). Compared to related works, this paper looses the assumption of independent arms distributions, Protocol 1 can estimate the means and covariances of arms and accelerate the best arm identification by utilizing the extra information. They propose two algorithms, one for scenario where the arms are bounded, and one for arms follow a Gaussian distribution. The lower bounds for both algorithms are also provided.

**Strengths:**

The idea of exploring the underlying structure of arms by allowing simultaneous queries is attractive. It not only solves the assumption of independent arms distributions, which is not realistic in various scenarios; but also accelerates best arm identification by leveraging the shared information between arms. Based on multiple queries protocol, the authors provide two main theorems for bounded variables and Gaussian distributions along with the corresponded lower bounds. They also manage to notice some failure mode of the algorithm, like shown in line 183, and give some solution. In addition, the paper is well-structured and easy to follow.

**Weaknesses:**

The authors emphasize their results are adaptive to unknown covariance, but based on the algorithms and simulations, it can only solve the type of same covariance between all arms (correct me if I were wrong), which reduces the practical use of this paper. In addition, as stated in the related work part, the topic of best arm identification under PAC-Learning framework is well-studied, so as the stochastic combinatorial semi-bandit problem. The motivation and main contributions of this papers are a little bit unclear. As for the simulation part, It would be helpful to provide some explanation for the three chosen benchmark algorithms to add on reliability. I was expecting to see the comparison of proposed algorithm and old algorithms who assume independent arms on scenarios where the assumption fails.

**Questions:**

1. Under the PAC framework, except for the confidence input $\delta$, the subtle difference $\epsilon$ is also important and related to the sample complexity, but I didn't find the involvement of this parameter in algorithms (like Theorem 4.1, line198).
2. If the assumption of arm independence holds, can we still use the proposed algorithms? Would there be a trade-off between complexity and generalization?
3. Could you provide a concrete example of the failure case mentioned in line 183? And how probably will this kind of failure happen?

**Limitations:**

This paper trying to utilize the covariances between arms to accelerate best arm identification, what if we step further, using the causal relationship instead of association? I know some research have been done in this area (like Lattimore, 2016) It would be interesting to consider this as a future direction. The proposed algorithms, although allow the dependent arms distributions, still have many constraints, like limited type of variables, only suitable for same covariance, pairwise comparisons, which reduce the practicality of this paper.

---

> ### Author Rebuttal · Authors · 2023-08-09
>
> We thank the reviewer for the valuable feedback.
> * **About the reviewer's summary and Question 1:** Please note that our objective in this paper is not identifying $\epsilon$-optimal arms (also known as $(\epsilon,\delta)$-PAC setting) but identifying the (exact) best arm with probability at least $1-\delta$.
> * **About the first point raised in the weaknesses section:** It is unclear to us what the reviewer suggests by "it can only solve the same covariance between all arms (correct me if I were wrong)". We would like to emphasize that we make no structural assumptions on the covariance between arms and our guarantees hold for any arbitrary covariance matrix of the joint distribution of the arms variables. In the bounded case, of course, the possible covariance matrix is implicitly constrained by the boundedness assumption.
> * **About Question 2:** In case the arms are independent our guarantees recover the standard known sample complexity up to a constant factor. However, whenever the arms variables have a strong correlation, our guarantees show that a significant improvement is made.
> * **About Question 3:** The failure case mentioned in line 183 may occur in some very specific scenarios. To illustrate consider the following toy example with $3$ arms: $X_1 \sim \mathcal{N}(\epsilon+\epsilon^{3/2},1)$, $X_2 = X_1-Z_2$ where $Z_2 \sim \mathcal{N}(\epsilon, \epsilon)$ and $X_3 = X_2-Z_3$ where $Z_3\sim  \mathcal{N}(\epsilon^{3/2}, (\epsilon/\kappa)^2)$, $\kappa>1$ is a constant to be specified and $X_1, Z_2$ and $Z_3$ are independent. Denote $\Lambda_{ij} = \frac{\text{Var}(X_i-X_j)}{\mathbb{E}[X_i-X_j]^2}$. Hence, in the previous example: $\Lambda_{12} = 1/\epsilon$, $\Lambda_{23} = 1/(\kappa^2\epsilon)$ and $\Lambda_{13} = (\epsilon+(\epsilon/\kappa)^2)/(\epsilon+\epsilon^{3/2})^2$. For $\epsilon$ very small we have $\Lambda_{12}, \Lambda_{13} \sim 1/\epsilon$ and $\Lambda_{23} \sim 1/(\kappa^2 \epsilon)$. Observe that the stopping time for comparing arms $i$ and $j$ (based on the test we considered) is a random variable $\tau$. We showed (Lemma E.4) that with high probability it belongs to the interval $[c_{\inf}\log(1/\delta_{\tau}) \Lambda_{ij}, c_{\sup}\log(1/\delta_{\tau})\Lambda_{ij}]$ (where $c_{\inf}<c_{\sup}$ are numerical constants). Therefore, in this example, arm $2$ eliminates arm $3$ faster than arm $1$. However, we may have that arm $1$ eliminates arm $2$ first at round $\sim c_{\inf} \Lambda_{12}$ (this can happen if we choose $\kappa^2 < c_{\sup}/c_{\inf}$), then we need to compare arm $1$ with arm $3$ paying a cost higher in average than the cost of comparing arms $2$ and $3$.

---

### Official Review · Reviewer_THM3 · 2023-07-12

**Soundness:** 3 good
**Presentation:** 2 fair
**Contribution:** 2 fair
**Rating:** 5
**Confidence:** 5

**Summary:**

This study addresses the problem of best arm identification (BAI) in the context of dependent arms. Unlike conventional settings, efficiency can be enhanced by exploiting the inherent correlation structure. This setting holds broad applications, including in clinical trials. Specifically, the authors concentrate on bandit scenarios with bounded and Gaussian rewards. The validity of their approach is substantiated through simulation studies.

**Strengths:**

The authors introduce a novel setting for Best Arm Identification (BAI) with fixed confidence. By accommodating correlation, we can devise more efficient strategies for identifying the optimal arm. This setting is intriguing and holds significant practical utility.

**Weaknesses:**

I believe several claims are not sufficiently substantiated. I have detailed these claims in the subsequent 'Questions' section. If the authors fail to adequately address my queries, these unsupported claims would constitute a weakness in the study.


**Questions:**

- In line 81, what does $\epsilon\mathcal{N}(1, 1)$ denote?
- In line 83, the authors suggest that since two queries are possible, the learner can perform a $T$-test. However, even in standard BAI, a setting based on such a hypothesis test, as suggested by the authors, exists. For instance, refer to Balsubramani & Ramdas (2016).
- Therefore, considering a hypothesis test in itself is feasible even in the standard setting. I believe that the key here is the variance reduction effect facilitated by two queries.
- In lines 84-86, the authors present an example of two-armed Gaussian bandits. They argue that the lower bound in standard BAI is $O((1+\epsilon)^2\log (1/\delta) / \epsilon^2)$. This seems to be a lower bound for the one-armed bandit problem. As highlighted by Kaufmann et al. (2016), the lower bound for two-armed Gaussian bandits is given as $O(Var(X_1 - X_2)) \log (1/\delta) / \epsilon^2)$ by assuming there are two arms, $Y_1 \sim N(\epsilon, Var(X_1 - X_2))$ and $Y_2 \sim N(0, 0)$. As I stated earlier, the essence of this example is not two queries, but treating two arms as one and comparing this combined arm to an arm that returns zero.
- For the related work section, I suggest considering the literature on BAI on graphs as it also deals with similar problems. Although the authors cite a few studies in this field, could these studies be related to yours?
- In line 125, the authors state that "BAI in the fixed confidence setting was studied by [8], [19], and [9], where the objective is to find $\epsilon$-optimal arms under the PAC model." Fixed confidence BAI is not confined to $\epsilon$-best arm identification. For instance, [17] considers a different objective.
- Subsequently, the authors claim that "A summary of various optimal bounds for this problem is presented in [5, 17]." Here, [5] considers lower bounds in BAI with a fixed budget, which is a different setting. [17] addresses the fixed-confidence BAI problem, but the objective is not $\epsilon$-optimal arms.
- In addition to the related work raised by the authors, it seems that Kato and Ariu (2021) have explored BAI with dependent arms for more specific cases. For example, they consider two-armed Gaussian bandits where rewards are correlated via contextual information. How is their study relevant to yours?

Balsubramani and Ramdas, Sequential nonparametric testing with the law of the iterated logarithm, UAI 2016.
Kato and Ariu, The Role of Contextual Information in Best Arm Identification, 2021.

**Limitations:**

None.

---

> ### Author Rebuttal · Authors · 2023-08-09
>
> We thank the reviewer for the valuable feedback.
> * **On Question 1**: The notation $\epsilon \mathcal{N}(1,1)$ stands for $\epsilon X$, where $X \sim \mathcal{N}(1,1)$.  We can write $\mathcal{N}(\epsilon,\epsilon^2)$ if it appears clearer, though we wanted to emphasize $\epsilon$ as a scaling factor.
> * **On Questions 2 and 3**: We absolutely agree that there are indeed many existing approaches based on arm elimination, which are all akin to performing sequential tests. What we want to emphasize here is that with 2 simultaneous queries, conceptually we can use a _paired samples_ $t$-test which is not possible in standard BAI. As noted by the reviewer, this allows for variance reduction. The work by Balsubramanu and Ramdas (2016) focuses on sequential hypothesis testing to make decisions between two possibilities. In their setup, they assume independent samples and utilize the empirical Bernstein concentration inequality tailored for bounded variables to adapt to the variances of the two variables under consideration. Similar adaptive approaches have been explored in other studies, and achieving adaptivity to arm variances is not a novel concept, as highlighted in our related work discussion. However, our main contribution lies in addressing the multiple arms case with arbitrary dependence. In this scenario, as explained in the introductory discussion (lines 78-99), certain intuitions from the standard setting no longer hold. For instance, it is possible that a sub-optimal arm may be eliminated by another sub-optimal arm more rapidly than by the optimal arm, which presents new challenges when dealing with multiple arms. Our approach is designed to tackle this situation and provide a solution in such settings. We will add a discussion in the related work section on Balsubramanu and Ramdas (2016) highlighting the common points of our procedure to theirs.
> * **On Question 4 (part 1)**: The 2-armed bandits case discussed in lines 78-99 serves as an illustrative example and does not represent a stand-alone contribution. Our primary objective is to address the more general $K$-armed bandit case. It is crucial to note that in the standard Best Arm Identification (BAI) setting, only one query per round is allowed. Therefore, the reduction involving comparing the difference arm "$X_1-X_2$" and $0$ is not feasible, as the learner lacks access to samples of the variable $Y_t = X_{1,t}-X_{2,t}$ directly. Instead, the learner has access to samples from $Y' \sim X_{1,t}-X_{2,s}$, where $s\neq t$ (resulting in $X_{1,t}$ and $X_{2,s}$ being independent). In this context, it is important to observe that $\text{Var}(Y') = \text{Var}(X_{1,t}-X_{2,s}) = \text{Var}(X_1) + \text{Var}(X_2)$, which can be much larger than $\text{Var}(Y) = \text{Var}(X_{1,t}-X_{2,t}) = \text{Var}(X_1-X_2)$. Achieving adaptivity to the latter variance is only possible with two queries per round, which is a crucial component for our results in addition to treating two arms as one in the 2-armed setting. In summary, being adaptive to the last variance is only possible with two queries per round, which is an important ingredient for our results besides treating two arms as one in the 2 armed setting.
> * **On Question 4 (part 2)**: Concerning the lower bound in the 2-arm setting: we are uncertain what the reviewer means with the 'one-arm bandit problem'. At any rate, reducing the (full) observation of the two arms $(X_1,X_2)$ to that of $(X_1-X_2, 0)$ results in a loss of information. Thus, a lower bound for the latter reduced setting does not (at least not without additional arguments) logically entail a lower bound for the initial setting. This is, however, somewhat beside the point we want to make, which is that
>  if we adhere to the standard BAI setting with one query per round, Kaufmann's lower bound is applicable and yields a sample complexity of $\mathcal{O}(\log(1/\delta)/\epsilon^2)$. However, it is essential to note that this lower bound does not extend to the multiple query case. In our work, we have presented a new lower bound specifically addressing the multiple query scenario in the $K$-armed bandits setting (Section 6).
> * **On Question 5: Bandits on graphs**: Previous studies on graph-based bandit problems with side observations in the stochastic setting (such as Caron et al., 2012) also explore the potential of simultaneously observing rewards from multiple arms. However, in these investigations, it is assumed that the distributions of arms are independent. We are not aware of any existing research on bandit problems in graph-based scenarios where the learner successfully attains adaptability to covariance among arms.
> * **On Question 8**: The study by Kato and Ariu (2021) examines the scenario where arms are correlated through contextual information. The primary distinction between their setting and ours lies in how the dependency between arms is accessed. In their work, the dependency between arms is accessible only through the context variable $X$, whereas in our approach, we directly obtain samples of the correlations between candidate's arms by jointly querying two arms. Additionally, another notable difference is in the context samples $X_t$ received at each round $t$. In their work, these context samples are independent of the arms chosen. However, in our case, the "observed dependency" between candidate arms is directly related to the specific arms queried in each round, allowing us to have direct access to correlations among the queried arms.
>
> Caron, Stéphane, Branislav Kveton, Marc Lelarge, and Smriti Bhagat. “Leveraging side
> observations in stochastic bandits.” UAI 2012.

---

> ### Comment · Area_Chair_RBeP · 2023-08-18
>
> Dear Reviewer,
>
> Please reply to the rebuttal and indicate whether it clears your concerns, or at least acknowledge whether you have read the response. This is important to the authors.
>
> Thanks,
>
> Your Area Chair

---

### Author Rebuttal · Authors · 2023-08-09

We thank the reviewers for the valuable feedback. We address below some points raised by the reviewers:

* **Link with bandits literature with dependent arms:**
     + **Bandits on graphs:** Previous studies on graph-based bandit problems with side observations in the stochastic setting (such as Caron et al., 2012) also explore the potential of simultaneously observing rewards from multiple arms. However, in these investigations, it is assumed that the distributions of arms are independent. We are not aware of any existing research on bandit problems in graph-based scenarios where the learner successfully attains adaptability to covariance among arms.
     + **Combinatorial semi-bandits:** about comparison with existing covariance-adaptive bandits works. While covariance-adaptive approaches have been explored in the context of combinatorial semi-bandits, it is important to note that the setting and objectives of this class of problems differ from ours. In our setting (Protocol 1), the learner has the flexibility to query any subset of arms, and the ultimate goal is to identify the single best arm. On the other hand, in combinatorial semi-bandits, the learner selects a subset from a predetermined set of subsets $\mathcal{M} = \\{ M_1, \dots, M_d \\}$ at each round $t$, where $M_i \subset \[K\]$ and $\lVert M_i\rVert_1 \le m$ for a specific problem parameter $m$. However, the objective in the latter case is not to identify the best arm individually but rather to find the subset $M_i$ with the largest sum of rewards. Given these differences, we refrained from comparing the guarantees obtained in our work with those of combinatorial semi-bandits. The dissimilarities in problem structures and primary objectives lead us to believe that such a comparison would not be meaningful.
* **On extending the assumptions to include sub-Gaussian variables:** We would like to clarify that our algorithm relies on the empirical Bernstein inequality, which was originally designed in the literature for bounded variables. However, we have extended this inequality to accommodate Gaussian variables by leveraging existing concentration results. It is essential to note that developing such inequalities for sub-Gaussian variables is generally a non-trivial task. One possible direction to extend the considered class of distributions is to suppose that arms follow a sub-Gaussian distribution and satisfy a Bernstein moment assumption (such extensions were pointed by works on bounded variables e.g., Balsubramani and Ramdas 2016). Given the last class of function, we can build on the standard Bernstein inequality with known variance, then plug in an estimate of the empirical variance leveraging the concentration of quadratic forms (see Bellec 2019). However, it remains uncertain whether an extension for sub-Gaussian variables (without additional assumptions) is practically feasible.
* **Position of our contributions with respect to the standard bandits setting and variance adaptive A/B testing:** Please refer to the second and third point of the rebuttal on reviewer THM3 for a more detailed discussion. In the standard BAI setting (one query per round) previous works (Mnih et al 2008, see our related work section) developed strategies that are adaptive to the variances of arms. In our setting, by allowing simultaneous queries we developed strategies that are adaptive to the variances of arms and covariances between arms. As shown by our introductory toy examples in lines 78-107, some basic intuitions in the standard setting no longer hold: for instance, we show that a sub-optimal arm may be eliminated by another sub-optimal arm much faster than by the optimal arm. Our main contribution is to develop strategies taking into account such observations among others, leading to guarantees that improve substantially the guarantees of the algorithms developed in the standard setting.
* **Regarding the cost of not knowing the variances:** Please refer to the second point of the rebuttal on reviewer pLAq for a more precise discussion. Regarding the overhead cost of unknown covariances, we acknowledge that in the scenario of independent arms, there is indeed a cost associated with being adaptive to covariances. However, we agree that this cost is merely a numerical factor when compared to the sample complexity of algorithms designed explicitly for independent arms. On the other hand, when arms are dependent, our guarantee demonstrates that being adaptive to covariance can result in a substantial improvement, possibly by an arbitrary factor, in certain scenarios. This improvement highlights the significance of considering covariance information in such situations and justifies the usefulness of our approach in dealing with dependent arms.

Volodymyr Mnih, Csaba Szepesvári, and Jean-Yves Audibert. Empirical Bernstein stopping. ICML 2008.

Balsubramani and Ramdas, Sequential nonparametric testing with the law of the iterated logarithm, UAI 2016.

P. C. Bellec. Concentration of quadratic forms under a Bernstein moment assumption. arXiv preprint 2019.

Caron, Stéphane, Branislav Kveton, Marc Lelarge, and Smriti Bhagat. “Leveraging side observations in stochastic bandits.” UAI 2012.

---

### Decision · Program_Chairs · 2023-09-21

**Decision:**

Accept (poster)

**Comment:**

This paper examines the best arm identification problem with a fixed confidence level, where arm distributions are dependent and multiple queries are allowed. The problem under study is both innovative and intriguing as it extends the practical application of Best Arm Identification methods in real-world scenarios. The authors present algorithms capable of discerning the exact best arm in this context, and provide both an upper and lower bound for sample complexity, demonstrating the optimality of the algorithm.

The paper has received unanimous acceptance recommendations. In accordance with the reviewers' viewpoints, I encourage the authors to incorporate all the points raised during the discussions into their final, camera-ready version of the paper. Congratulations on this commendable work.